# Beyond ReLU: Bifurcation, Oversmoothing, and Topological Priors

Erkan Turan [1]   Gaspard Abel [2 3]   Maysam Behmanesh [1]   Emery Pierson [1]   Maks Ovsjanikov [1]

## Abstract

Graph Neural Networks (GNNs) learn node representations through iterative network-based message-passing. While powerful, deep GNNs suffer from oversmoothing, where node features converge to a homogeneous, non-informative state. We re-frame this problem of representational collapse from a *bifurcation theory* perspective, characterizing oversmoothing as convergence to a stable "homogeneous fixed point." Our central contribution is the theoretical discovery that this undesired stability can be broken by replacing standard monotone activations (e.g., ReLU) with a class of functions. Using Lyapunov-Schmidt reduction, we analytically prove that this substitution induces a bifurcation that destabilizes the homogeneous state and creates a new pair of stable, non-homogeneous *patterns* that provably resist oversmoothing. Our theory predicts a precise, nontrivial scaling law for the amplitude of these emergent patterns, which we quantitatively validate in experiments. Finally, we demonstrate the practical utility of our theory by deriving a closed-form, bifurcation-aware initialization and showing its utility in real benchmark experiments.

## 1. Introduction

Graph Neural Networks (GNNs) have emerged as a powerful class of models for relational data, with architectures like Graph Convolutional Networks and Graph Attention Networks achieving strong results in many applications (Kipf & Welling, 2017; Veličković et al., 2018; Hamilton et al., 2017). Despite this promise, GNNs suffer from a fundamental limitation known as *oversmoothing*: as network depth increases, node features converge to increasingly similar representations (Li et al., 2018; Oono & Suzuki, 2020; Chen et al., 2020). In practice, this restricts most architectures to just 2–4 layers, a stark contrast to the very deep networks that have proven effective for images and text. This depth limitation is particularly problematic for tasks requiring long-range communication, since most GNN architectures exchange information only between neighboring nodes (Alon & Yahav, 2021).

The research community has invested considerable effort into both practical mitigations and theoretical understanding of oversmoothing (Rusch et al., 2023). Practical solutions include residual connections (Li et al., 2019) and normalization techniques (Zhao & Akoglu, 2020), while theoretical analyses have drawn on spectral theory (Oono & Suzuki, 2020) and, more recently, dynamical systems perspectives such as contractive mappings (Arroyo et al., 2025) and ergodic theory (Zhao et al., 2025). However, existing theoretical investigations place little emphasis on the role of activation functions, and when they do, the analysis is typically restricted to standard choices such as ReLU. A separate line of work has focused on continuous-time formulations (Chamberlain et al., 2021; Behmanesh et al., 2023), viewing GNN updates as discretizations of underlying differential equations and drawing on reaction–diffusion models from physics (Choi et al., 2023).

In this work [1], we step back from continuous-time variants and study classical discrete message-passing architectures. We investigate oversmoothing through the lens of bifurcation theory, revealing a previously unappreciated connection to the choice of activation function coupled with the role of the weight initialization during training. Motivated by our analytical derivations, we propose a simple fix: substituting the common ReLU activation with a class of activations our theory identifies, odd functions with a stabilizing cubic term, such as the non-monotonic Sine. This substitution alone, with our derived bifurcation-aware initialization scheme, provably prevents collapse to the trivial fixed point, destabilizing the homogeneous state and stabilizing informative, non-homogeneous fixed points. A second, independent lever, polynomial spectral filtering, then selects which topological mode this prior aligns with, yielding competitive

---

[1]LIX, Ecole Polytechnique, IP Paris [2]Université Paris Saclay, Université Paris Cité, ENS Paris Saclay, CNRS, SSA, INSERM, Centre Borelli, F-91190, Gif-sur- Yvette, France [3]Centre d'Analyse et de Mathématique Sociales, EHESS, CNRS, 75006 Paris, France.. Correspondence to: Erkan Turan <turan@lix.polytechnique.fr>.

*Proceedings of the 43$^{rd}$ International Conference on Machine Learning*, Seoul, South Korea. PMLR 306, 2026. Copyright 2026 by the author(s).

[1]The code used for this paper can be found here: https://github.com/maysambehmanesh/relu-bifurcation

performance against state-of-the-art GNN architectures.

Our contributions are as follows:

- **Bifurcation analysis of activation functions.** We establish that oversmoothing can be overcome by suitable classes of activation functions (Theorem 3.2). We show that the choice of activation instantiates a particular dynamical system and characterize function classes that promote bifurcating behavior (pitchfork or transcritical). We rigorously prove that these activations destabilize the homogeneous fixed point and derive non-trivial, testable scaling laws for the amplitude of emergent patterns.

- **Graph-aware initialization.** By combining our bifurcation analysis with random matrix theory, we derive closed-form initialization formulas that position GNNs precisely at the critical point (Corollary 4.2). This provides a principled, graph-aware alternative to standard variance-preserving initialization schemes.

- **Empirical validation.** Through experiments on both a controlled toy model and realistic GNN architectures, we validate our theoretical predictions for pattern amplitude scaling across diverse graph topologies. On node classification benchmarks, 64-layer GNNs initialized according to our scheme achieve performance, with phase transitions occurring near the theoretically predicted parameter values.

In summary, our bifurcation-theoretic perspective reveals that oversmoothing is not inevitable, but rather a consequence of specific dynamical properties that can be engineered away through principled activation and initialization choices.

## 2. Related Work

**Oversmoothing as a spectral property in GNNs.** The phenomenon of oversmoothing was theoretically investigated by Li et al. (2018); Oono & Suzuki (2020); Chen et al. (2020), who showed that deep GNNs lose discriminative power as node features converge. From a spectral view, message-passing is usually described as a filter (NT & Maehara, 2019; Wu et al., 2019), progressively smoothing features toward the graph's dominant eigenvector. Recent work formulate this collapse towards homogeneous states using operator semi-group theory (Zhao et al., 2025), or vanishing gradients (Arroyo et al., 2025), and provide results on its exponential convergence. In parallel, a multitude of architectural solutions have been developed to mitigate oversmoothing in GNNs, including residual connections (Scholkemper et al., 2025; Li et al., 2019), normalization (Zhao & Akoglu, 2020) and graph rewiring (Topping et al., 2022; Karhadkar et al., 2023).

**A physical perspective on GNNs.** Recent work has also drawn inspiration from physics to address oversmoothing. An energy-based formulation of iterative graph convolutions was used in (Giovanni et al., 2023) to analyze GNN convergence properties, as well as a dynamical system approach in their attention-based variants in Wu et al. (2023). The continuous-time limit formulation of GNN layers has given birth to a multitude of GNN architectures: GREAD (Choi et al., 2023), ACMP (Wang et al., 2023) and RDGNN (Eliasof et al., 2024) formulate GNNs as reaction-diffusion PDEs, to enable pattern formation through learnable reaction terms. Other architectures such as (GraphCON) Rusch et al. (2022), KuramotoGNN Nguyen et al. (2024) and Stuart-Landau GNN (Zhang et al., 2026) model node features as a system of nonlinear coupled oscillators, demonstrating that oscillatory dynamics can prevent the convergence to trivial steady states. Complementary to such dedicated architectures, we provide bifurcation-theoretic foundations to rigorously examine GNN design and study the emergence of these expressive patterns.

**Activation functions and initialization.** Despite its well-established development in computer vision with PReLU (He et al., 2015) or SIREN (Sitzmann et al., 2020) for implicit neural representations, the role of activations in GNNs has not been thoroughly investigated beyond empirical tests (Rasool et al., 2024). We provide a theoretical basis for activation selection based on dynamical stability and rigorously characterize which properties of activation functions induce stable pattern-forming bifurcations. For initialization, while recent graph-oriented methods (Li et al., 2023) use variance analysis, our bifurcation-aware approach initializes GNNs at critical points which, as we show, combined with topological mode selection (Jacot et al., 2018), instantiate GNN training towards expressive representations.

## 3. A Dynamical System Perspective for Oversmoothing

### 3.1. Oversmoothing as Fixed-Point Convergence

Oversmoothing in GNNs is commonly interpreted through a spectral lens: repeated message passing acts as a low-pass filter on node features (NT & Maehara, 2019; Wu et al., 2019). More recently, this phenomenon has been reframed from a dynamical system perspective (Zhao et al., 2025; Arroyo et al., 2025). In this view, a deep GNN defines a discrete dynamical system

$$x^{(\ell+1)} = f(x^{(\ell)}), \qquad (1)$$

where oversmoothing corresponds to convergence toward a stable *homogeneous fixed point* of the layer-wise node representations $(x^{(\ell)})_{\ell=1,...L}$. The Banach fixed-point theorem ensures the convergence of the system to a unique attractor.

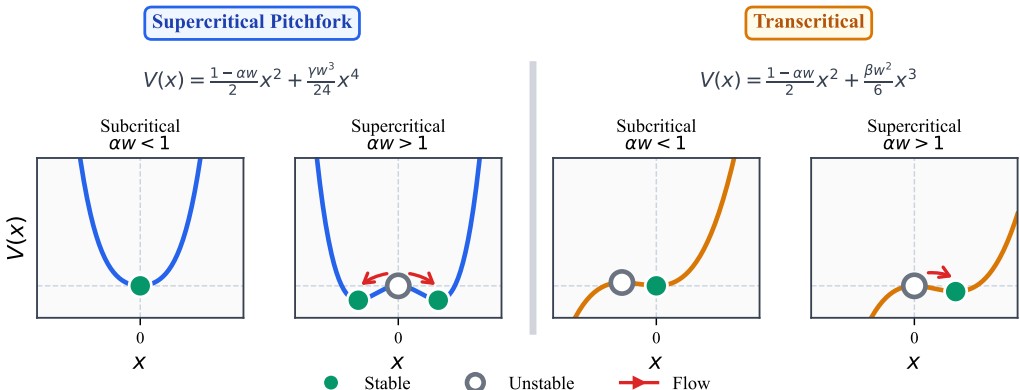

*Figure 1.* **Effective potential landscape across bifurcation.** *Left:* Supercritical Pitchfork bifurcation. *Right:* Transcritical bifurcation. For both bifurcation types, below the critical coupling $w < 1$, the system is at the subcritical regime – the homogeneous state $a = 0$ is the unique stable minimum, corresponding to oversmoothing. Above $w > 1$, the origin becomes unstable (hollow circle) and stable minima emerge (filled circles), representing non-homogeneous pattern formation that resists oversmoothing.

**Lemma 3.1.** *Let $f$ be an operator with Lipschitz constant $\|f\|_{\mathrm{Lip}} < 1$. Then, for any initial condition $x_0$, iteration $x^{(\ell+1)} = f(x^{(\ell)})$ converges linearly to the unique fixed point of $f$.*

For GNN architectures with standard activations (e.g. ReLU) oversmoothing can be seen as a contractivity issue (Arroyo et al., 2025): the iterative layer map $f$ causes the representations to converge toward the zero vector, making oversmoothing inevitable under contractivity. This suggests a natural question: what happens if we break contractivity? Without the guaranty of a unique attractor, the system may admit multiple fixed points that can be non-trivial. Indeed, we will show that equipped with suitable activation functions, the homogeneous fixed point of the GNN can be destabilized and create new stable attractors that encode meaningful information about the graph topology. This perspective reframes oversmoothing as a problem of *stability engineering*, which will be examined through the lens of bifurcation theory.

### 3.2. Bifurcation Theory and Stability Engineering

The previous section established that oversmoothing arises because standard GNNs converge to a trivial fixed point. To overcome this, we need the system to support non-trivial stable equilibria. Our key observation is:

> *The choice of activation function determines the topology of the stability landscape.*

To make this precise, consider the simplified GNN layer update on a single node graph (no connectivity matrix for now), parametrized by a coupling strength $w$ and general activation function $\phi$ (interpreted as a weight):

$$x^{(\ell+1)} = \phi(wx^{(\ell)}), \qquad (2)$$

**Fixed points as energy minima** Fixed points satisfy $\phi(wx^{(\ell)}) - x^{(\ell+1)} = 0$. This condition can be locally

viewed as the extremum of an *effective potential $V(x)$* defined via

$$-\frac{dV}{dx} = \phi(wx) - x, \quad \text{i.e.,} \quad V(x) = \int [x - \phi(wx)]\, dx. \qquad (3)$$

This formulation provides a powerful geometric intuition: fixed points correspond to stationary points of $V$, with *stable* fixed points at local minima and *unstable* ones at local maxima, much like a ball rolling on a landscape comes to rest only at the bottom of valleys (Figure 1). More details of this paradigm are provided in Appendix A.1.

**How the activation shapes the stability landscape** The form of $\phi$ directly determines the shape of $V(x)$, and hence the number and stability of fixed points. To see this, expand a smooth activation around the origin.

$$\phi(x) \approx \alpha x - \frac{\beta}{2}x^2 - \frac{\gamma}{6}x^3 + \dots \qquad (4)$$

Substituting in the potential yields

$$V(x) = \frac{1 - \alpha w}{2}x^2 + \frac{\beta w^2}{6}x^3 + \frac{\gamma w^3}{24}x^4 + O(x^6). \quad (5)$$

The coefficients $\alpha = \phi'(0), \beta = \phi''(0)$ and $\gamma = -\phi'''(0)$ determine the landscape topology.

- **Quadratic term** $(1 - \alpha w)/2$: determines whether $x = 0$ is a minimum (positive) or maximum (negative).

- **Quartic term** $\gamma w^3/24$: if positive, it "walls off" the potential at large $|x|$, creating new minima when the origin becomes unstable.

**Classical examples.** Two canonical bifurcations are illustrated in Figure 1, to show how the coefficients $\alpha$, $\beta$, $\gamma$ shape the landscape:

- **Supercritical pitchfork** ($\beta = 0$, $\gamma > 0$): Odd activations like $\sin(z)$ or $\tanh(z)$ have vanishing quadratic term. The potential $V(x) = \frac{1-\alpha w}{2}x^2 + \frac{\gamma w^3}{24}x^4$ is symmetric: for $\alpha w > 1$, the single well splits into a double-well with two minima at $x^* = \pm\sqrt{6(\alpha w - 1)/\gamma w^3}$.

- **Transcritical** ($\beta > 0$, $\gamma = 0$): Activations with a non-zero quadratic term yield an asymmetric potential $V(x) = \frac{1-\alpha w}{2}x^2 + \frac{\beta w^2}{6}x^3$. For $\alpha w > 1$, a single new stable minimum emerges at $x^* \propto (\alpha w - 1)$.

In both cases, crossing the critical threshold $\alpha w = 1$ destabilizes the origin and creates non-trivial stable states, precisely the mechanism needed to escape oversmoothing.

**Two contrasting activations.** Consider $\phi(z) = \sin(z)$ versus $\phi(z) = \mathrm{ReLU}(z)$:

- **Sine:** $\alpha = 1$, $\gamma = 1 > 0$. For $w < 1$, the potential is a single well at $x = 0$. For $w > 1$, the quadratic coefficient flips sign, but the positive quartic term creates a double-well with two new stable minima at $x^* = \pm\sqrt{6(w-1)/w^3}$.

- **ReLU:** $\alpha = 1$ (for $x > 0$), but $\gamma = 0$: there is no quartic restoring force. For $w > 1$, contractivity is broken, which causes the potential to tilt downward indefinitely. Consequently, iterations will diverge rather than settle into new stable states.

This observation poses a fundamental design question: can we, through a specific choice of the activation function, re-engineer the GNN's dynamics to destabilize this homogeneous fixed state, in favor of creating *stable, and informative non-homogeneous patterns*? The next section presenting our main theoretical results provides an affirmative answer.

### 3.3. Activations functions as Bifurcating Systems
We now add connectivity between nodes to investigate the role of topology in the GNN dynamics.

$$x^{(\ell+1)} = \phi(wAx^{(\ell)}) \tag{6}$$

where $x^{(\ell)} \in \mathbb{R}^N$ is one-dimensional node representations, $w$ is a scalar coupling parameter and $A \in \mathbb{R}^{N \times N}$ is the normalized adjacency matrix. Despite its simplicity, this update captures the core structure in widely-used architectures like GCN and GAT, where node representations are iteratively aggregated over the graph and passed through a point-wise nonlinearity. This minimal setting allows us to extract the essential mechanism by which activation functions govern the stability landscape, insights we generalize to realistic multi-feature GNNs in Section 5.

While most oversmoothing mitigation strategies append components to this architecture or stray away from it, our main theoretical result demonstrates that with an appropriate choice of activation function, increasing the coupling $w$ destabilizes the trivial homogeneous solution $x = 0$ triggering a bifurcation that creates a new pair of stable, non homogeneous fixed points.

**Theorem 3.2** (Supercritical pitchfork from odd $C^3$ activations). *Let $A \in \mathbb{R}^{N \times N}$ be the normalized symmetric graph adjacency matrix with eigenpairs $Au_j = \lambda_j u_j$, and assume $\lambda_k > 0$ is the simple maximum eigenvalue with unit eigenvector $u_k$. Consider the discrete dynamical system*

$$x^{(\ell+1)} = \phi(w A x^{(\ell)}), \tag{7}$$

*where $\phi \in C^3(\mathbb{R})$ acts entrywise and satisfies $\phi(0) = 0$. Assume $\phi$ is* odd *and*

$$\phi'(0) = \alpha > 0, \qquad \phi'''(0) = -\gamma < 0.$$

*Define the critical coupling $w_k := \frac{1}{\alpha\lambda_k}$. Then there exists $\varepsilon > 0$ such that for all $w \in (w_k, w_k + \varepsilon)$:*

1. *(**Existence and shape**) Equation (7) admits exactly two nonzero fixed points $x^\star_\pm(w)$ in a neighborhood of $0$, and they satisfy*

$$x^\star_\pm(w) = \pm a^\star(w)\, u_k + O\big((a^\star(w))^3\big).$$

2. *(**Square-root scaling**) Let $\mu := \alpha w \lambda_k - 1$, $\kappa_k := \sum_{i=1}^N u_{k,i}^4 > 0$. The amplitude obeys the supercritical pitchfork law*

$$a^\star(w) = \sqrt{\frac{6\,\mu}{\gamma\,(w\lambda_k)^3\,\kappa_k}} + O(\mu^{3/2}).$$

3. *(**Stability**) The bifurcated fixed points $x^\star_\pm(w)$ are locally exponentially stable, while the homogeneous fixed point $x = 0$ loses stability at $w = w_k$.*

The proof is given in Appendix A.4. This theorem reveals a critical interplay between the choice of the activation function, learnable parameters and graph topology $\alpha$, $w$, $(\lambda_k, u_k)$. It yields a concrete design principle to avoid oversmoothing, replace activations like ReLU with alternatives possessing (i) a positive slope at zero ($\alpha = \phi'(0) > 0$) and ii) a *stabilizing* cubic nonlinearity ($\gamma = \phi'''(0) < 0$). Common activations satisfying these conditions include $\sin(x)$ and $\tanh(x)$

*Remark* 3.3. This is not the only route to avoid oversmoothing. Activations with $\alpha > 0$ and $\phi''(0) < 0$ induce a *transcritical* bifurcation instead. Interestingly, the Allen–Cahn and Fisher reactions introduced in Choi et al. (2023) correspond to pitchfork and transcritical systems, respectively. However, their framework relies on continuous-time reaction-diffusion PDEs that mitigate oversmoothing, whereas our result applies to standard discrete message-passing with a simple activation substitution.

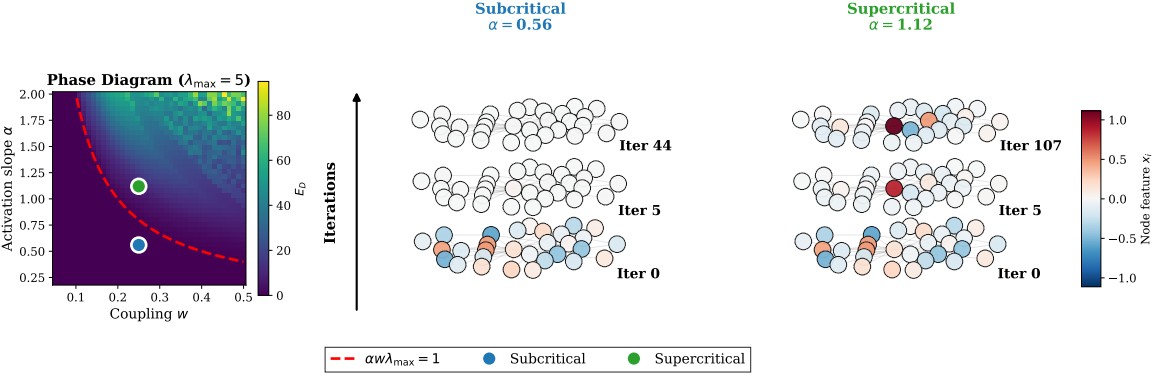

*Figure 2.* **Phase diagram validating Theorem 3.2.** *Left*: Dirichlet energy $E_D$ as a function of activation slope $\alpha$ and coupling $w$ (dashed red) precisely separates the oversmoothing regime $E_D \approx 0$ for the pattern forming regime $E_D > 0$. *Right*: Node representations at marked points. In the subcritical regime all representations collapse to zero (homogeneous fixed point). In the supercritical regime, a structured pattern emerges.

**Quantifying the departure from oversmoothing** This theorem proves that by modifying the activation function, a simple GNN model can have *new* stable fixed points that are non-zero ($a_\star > 0$) and non-homogeneous (aligned with the leading eigenvector $u_k$). These structured patterns resist oversmoothing by maintaining representational diversity across nodes. To quantify this effect, we derive the Dirichlet energy, a common measure of oversmoothing (Chen et al., 2020; Choi et al., 2023; Cai & Wang, 2020), in the following corollary :

**Corollary 3.4** (Dirichlet Energy). *Under the same assumptions as in Theorem 3.2, let $L$ be the associated normalized graph Laplacian, the Dirichlet energy $E_d(x) := x^\top L x$ of the fixed-point solutions satisfies:*

$$E_D^\star = E_D(x^\star) = \begin{cases} 0, & \mu \leq 0 \\ C_k \mu + O(\mu^2), & \mu > 0, \end{cases}$$

*where $\mu := \alpha w \lambda_k - 1$ is the bifurcation parameter and $C_k = \frac{6(1-\lambda_k)}{\gamma(w\lambda_k)^3 \kappa_k} > 0$ is a positive constant.*

The proof is given in A.5. Therefore, the Dirichlet energy remains zero in the subcritical (oversmoothing) regime but becomes positive and grows *linearly* with the bifurcation parameter $\mu$ in the supercritical regime (pattern formation).

The bifurcation condition $\alpha w \lambda_k = 1$ is empirically validated in Figure 2, which plots the Dirichlet Energy $E_D$ after iterating the mapping in Eq. 6 of randomly initialized representations to convergence on Barabási-Albert graphs (see Appendix for details).

**Connection to Effective Potential** This bifurcation mechanism admits an intuitive energy-landscape interpretation. The amplitude $a^*$ minimizes an effective potential analogous to the Landau free energy in phase transition theory.

**Corollary 3.5** (Effective Potential Energy). *The bifurcating*

*fixed-point amplitude $a_\star$ from Theorem 3.2 minimizes the "effective potential energy" $V(a)$:*

$$V(a) = \frac{(1 - \alpha w \lambda_k)}{2} a^2 - \gamma \frac{(w\lambda_k)^3}{4} \kappa_k a^4 + O(a^6).$$

The proof is given in A.6. This energy function corresponds to a *Supercritical Pitchfork* bifurcation system as in (see Fig.1) and elegantly explains the bifurcation mechanism:

- **Subcritical regime ($\alpha w \lambda_k < 1$):** The coefficient of the $a^2$ term $(1 - \alpha w \lambda_k)/2$ is positive. $V(a)$ is a single well with its minimum at $a = 0$. The system is trapped in the stable, homogeneous (oversmoothed) state.

- **Supercritical regime ($\alpha w \lambda_k > 1$):** The coefficient of the $a^2$ term becomes negative, destabilizing $a = 0$ into an unstable local *maximum*. The $a^4$ term (which is positive) bounds the energy, creating two new, symmetric minima at $a_\star \neq 0$.

**Empirical verification of scaling laws.** The theoretical predictions of this section yield precise, testable scaling laws : the amplitude grows as $a^\star \sim \sqrt{\mu}$ (Theorem 3.2) and the Dirichlet energy grows as $E_D^\star \sim \mu$ (Corollary 3.4). Although our derivation mode-coupling reduction is local, Figure 3 validates both predictions across Erdős–Rényi, Barabási–Albert, Watts–Strogatz, and random-regular graphs. In particular, the log-log plot (Figure 3d) confirms the predicted $\sqrt{\mu}$ scaling, with all graph types collapsing onto the theoretical curve near the bifurcation point $\mu = 0$.

## 4. Bifurcation Unlocking Topological Learning Priors

We have established that bifurcation prevents the collapse of node representations to a trivial state. We now show that this

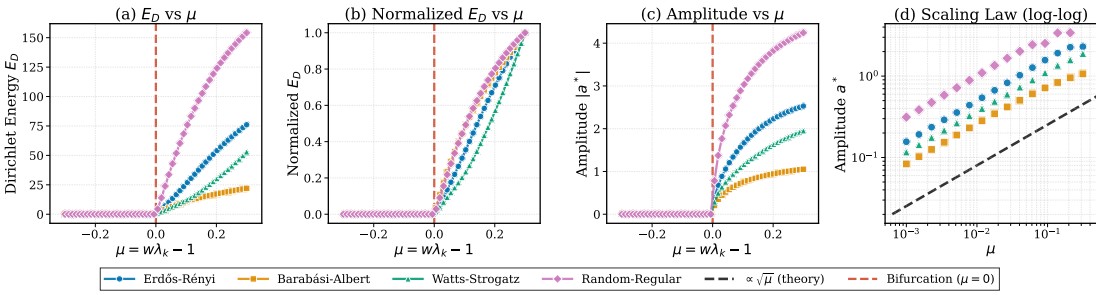

*Figure 3.* **Empirical validation of theoretical predictions.** Dirichlet energy, amplitude and scaling laws versus normalized coupling $\frac{w}{w_k}$ on Erdos-Renyi, Barabasi-Albert, Watts-Strogatz and Random-Regular graph topologies

mechanism plays a second, equally critical role: it induces a strong *inductive bias* that accelerates learning for specific topological patterns.

### 4.1. Neural Tangent Kernel Analysis of Bias in Learning Dynamics

This section shows that initializing near the bifurcation point does more than prevent collapse: it induces a strong *learning prior* that biases optimization toward a specific graph eigenmode. Using the Neural Tangent Kernel (NTK) framework (Jacot et al., 2018), we show that near criticality the kernel becomes asymptotically rank-one, causing gradient descent to preferentially learn patterns aligned with the bifurcating mode. More details are given in Appendix A.7.

Under gradient descent, the network output evolves as

$$\frac{d}{dt}(f_t - \mathcal{Y}) = -\eta K(f_t - \mathcal{Y}), \tag{8}$$

where $K$ is the NTK. Decomposing the error into eigenmodes of $K$ shows that modes associated with larger kernel eigenvalues are learned faster.

We apply this analysis to our scalar GNN model. At initialization, the prediction at node $i$ is the fixed-point value $f_w(i) := x_i^\star(w)$. Since the only learnable parameter is $w$, the node–node NTK reduces to

$$K_w(i,j) := \partial_w f_w(i)\, \partial_w f_w(j). \tag{9}$$

By analyzing the sensitivity of the bifurcating fixed point $x^\star(w)$ with respect to $w$, we obtain the following result.

**Corollary 4.1** (NTK Mode Selection at Bifurcation). *Let $x^\star(w)$ denote the initialized node features, $(u_r)_{r=1}^n$ the eigenvectors of the adjacency matrix $A$, and $\mu := \alpha w \lambda_k - 1$ the bifurcation parameter.*

*1. (Supercritical regime: $w > w_k$, $0 < \mu \ll 1$)*

$$K_w(i,j) = \frac{C}{\mu}\, u_k(i)\, u_k(j) + O(1),$$

*where $C > 0$ depends on $\alpha, \lambda_k, \gamma$, and $\kappa_k$. The kernel is asymptotically rank-one and dominated by the bifurcating mode $u_k$.*

*2. (Subcritical regime: $w < w_k$)*

$$K_w(i,j) \propto \sum_r \frac{1}{(1 - \alpha w \lambda_r)^2}\, u_r(i)\, u_r(j),$$

*with no diverging eigenvalue and no single selected mode.*

**Interpretation.** Below the bifurcation, the NTK spectrum is bounded and learning is distributed across graph modes. Near criticality, the kernel eigenvalue associated with $u_k$ diverges as $\lambda_k^{(K)} \propto 1/\mu$, implying that the learning time of this mode scales as $\tau_k \sim (\eta \lambda_k^{(K)})^{-1} \propto \mu$ and vanishes as $\mu \to 0$. Consequently, gradient descent is strongly biased toward the graph eigenmode selected by the bifurcation.

> *Criticality selects the graph mode; the kernel inherits it as a learning prior.*

### 4.2. Spectral Filtering as Prior Selection

By default, the bifurcation mechanism selects the eigenmode $u_k$ with largest eigenvalue $\lambda_k$, as it has the smallest critical threshold $\alpha w \lambda_k = 1$. However, different tasks benefit from different topological priors.

To control mode selection, we replace $A$ with a polynomial filter $P(A) = \sum_{j=0}^{K} \theta_j A^j$. The bifurcation condition becomes $\alpha w P(\lambda_k) = 1$, so the first mode to destabilize is $\arg\max_k P(\lambda_k)$. By shaping $P$, we directly control which prior emerges, to adapt to the task and the graph type. Specifically:

- *Homophilic graphs*: Low-pass filters select smooth eigenmodes aligned with clusters.

- *Heterophilic graphs*: High-pass filters select oscillatory modes distinguishing adjacent nodes.

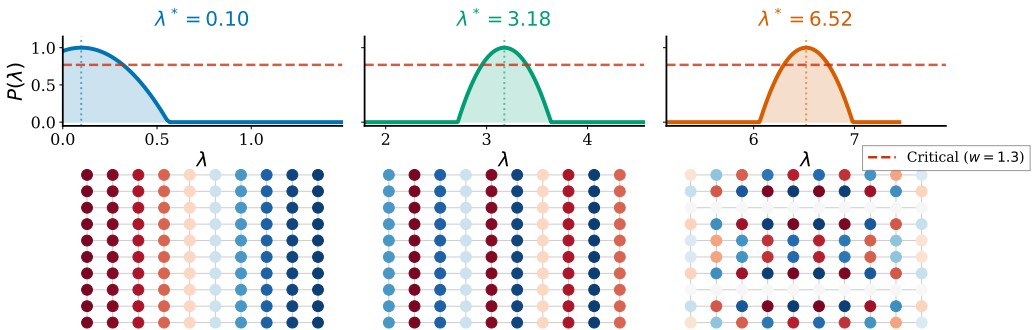

*Figure 4.* **Spectral filtering controls mode selection.** *Top*: Bandpass filters $P(\lambda)$ centered at low, mid and high frequencies. The dashed line is the critical threshold $\frac{1}{\alpha w}$, only modes that exceed this threshold can bifurcate. *Bottom*: Stable patterns on a $10 \times 10$ grid graph obtained by iterating $x^{(\ell+1)} = \phi(wP(A)x^{(\ell)})$ to convergence from random initializations. Each filter selects distinct eigenmode: smooth (low frequency), stripped (mid-frequency), or checkerboard (high-frequency), which illustrates topological prior selection without collapse.

Figure 4 demonstrates this on a grid graph, where filtered dynamics $x^{(\ell+1)} = \phi(wP(A)x^{(\ell)})$ converge to structured patterns determined the eigenmodes selected by the filter.

*Remark* 4.2. While polynomial filters appear in Cheb-Net (Tang et al., 2024) and GPR-GNN (Chien et al., 2021), our contribution is the bifurcation-theoretic interpretation: $P$ determines *which* mode is selected, while nonlinearity determines *whether* patterns emerge.

# 5. Generalization to Realistic GNN Architectures

The 1D model in Section 3.2 provides the core analytic mechanism. We now generalize this discovery to the realistic, multidimensional GNN setting. We analyze the discrete map corresponding to a GNN layer with $d$-dimensional features, where the graph operator $A$ acts on the nodes and a weight matrix $W$ mixes the features. For the sake of brevity, we formulate here Theorem 5.1 a simplified version of the main Theorem A.1 in the Appendix (followed by its proof in A.9).

**Theorem 5.1** (GNN Bifurcation on realistic GNNs). *Consider the GNN layer update $X^{(\ell+1)} = \phi(A\,X^{(\ell)}\,W)$ with normalized adjacency $A$, random weights $W$, and nonlinear activation $\phi$ satisfying $\phi'(0) = \alpha > 0$, $\phi'''(0) = -\gamma < 0$.*

*Let $\lambda_k$ and $\rho(W)$ be the largest eigenvalues of $A$ and $W$ respectively. Then:*

1. *The zero fixed point loses stability when $\alpha\rho(W)\,\lambda_k = 1$, triggering a pitchfork bifurcation.*

2. *For $\alpha\rho(W)\,\lambda_k > 1$, two stable equilibria emerge as rank-one patterns: $X_\pm^\star = \pm a^\star\,u_k v_j^\top +$ higher order terms, where $u_k$, $v_j$ are dominant eigenmodes of $A$, $W$.*

This extends our 1D analysis to realistic GNNs. The bifurca-

tion occurs when the graph's dominant frequency ($\lambda_k$) and feature mixing ($\rho(W)$) combine to destabilize uniformity. Emergent patterns are separable states $u_k v_j^\top$ that align with both graph structure and feature space, showing how GNN dynamics naturally discover structured representations.

## 5.1. A Bifurcation-Aware Initialization Scheme

The bifurcation condition $\alpha\rho(W)\,\lambda_k = 1$ yields a *practical initialization strategy*. Using random matrix theory, we can set the weight variance to place the GNN exactly at the critical point:

**Corollary 5.2** (Critical initialization). *For Ginibre-type weights, the critical variance is*

$$v_c \approx \frac{1}{d\lambda_k^2\alpha^2} \quad (10)$$

The proof is given in Appendix 5.2. This provides a simple recipe: compute $\lambda_k$ and use (10) to initialize at the bifurcation point. Unlike standard heuristic schemes that preserve signal variance, this "bifurcation-aware" initialization aligns with the system's stability landscape. (Li et al., 2023).

## 5.2. Learnable polynomial spectral filters

By Theorem 5.1, the activation substitution already guarantees escape from the trivial fixed point; the bifurcation then selects, by default, the dominant eigenmode of $A$ (typically low-frequency, per Corollary 4.1). This default prior suits homophilic tasks well (e.g. CORA, Figure 5), but others benefit from a different critical mode. We therefore add a second, independent lever that controls *which* mode survives, without altering the anti-collapse guaranty. Concretely, we replace the fixed operator $A$ with a learnable polynomial filter $P(A) = \sum_{k=0}^{K} c_k A^k$, whose trainable coefficients $\{c_k\}$ reshape the spectrum so that the task-relevant mode is the one that bifurcates (Figure 4). We pair this with

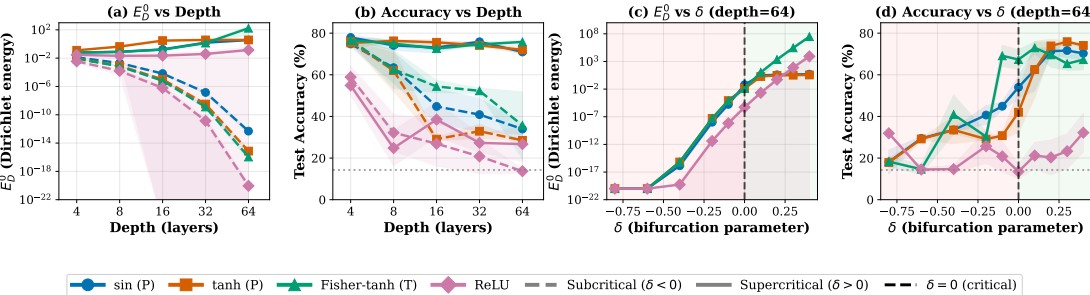

*Figure 5.* **Depth Robustness and Phase Transitions on CORA. (a, b)** $E_D^0$ and accuracy vs. depth. Activations with stabilizing cubic terms (sin, tanh) resist oversmoothing up to 64 layers, whereas ReLU collapses. **(c, d)** Phase transition at depth 64. Varying the bifurcation parameter $\delta$ reveals a transition near $\delta = 0$; only supercritical initialization ($\delta > 0$) enables pattern formation and successful learning.

a sine activation in two variants: *Sine-Poly(A)-sh*, which shares a single polynomial filter across all layers, and *Sine-Poly(A)-pl*, which uses independent per-layer filters with layer-specific coefficients $\{c_k^{(\ell)}\}$. Each layer applies the polynomial aggregation, a linear transformation, and a sine activation.

## 6. Experiments

Our experiments[2] are conducted using a standard baseline GCN architecture and consist of verifying the following statements:

- **Isolation test for oversmoothing:**: Using a node-classification task with vanilla GNNs at increasing depth, we show that the activation substitution alone, with our derived initialization, and without any spectral filtering, lets deep GNNs break oversmoothing, while ReLU collapses.

- **Node classification results:** We then add mode selection as a separate lever. Equipped with learnable polynomial mode selection, these architectures are competitive on node-classification benchmarks across all homophily levels. Mode selection does not affect the anti-collapse property; it only controls which topological mode the prior aligns with, and the proposed activations do not cripple expressivity in standard shallow tasks.

- **Ablation of activation functions:** Lastly, we ablate the construction of our best performing architecture – Sine-Poly(A) – by removing the polynomial filtering and changing the activation function. Our results demonstrate the superiority of the Sine activation function, validating the theoretical results, as well as the need for polynomial filtering in practical applications.

---

[2]The code used to reproduce the following results can be found here: https://github.com/maysambehmanesh/relu-bifurcation

### 6.1. Robustness to depth.

To validate the resistance of specific activations to over-smoothing, we test our bifurcation-aware initialization with a controlled variance $v_c(\delta) = \frac{1+\delta}{d\lambda_k^2\alpha^2}$ for depth robustness on the CORA dataset. Figure 5(a,b) shows that with super-critical initialization ($\delta > 0$), bifurcating activations (sin, tanh, Fischer-tanh) maintain high initial Dirichlet energy $E_D^0$ and stable accuracy up to 64 layers, while the latter collapses for ReLU. In Figure 5(c,d), with a 64-layer GNN, a parameter sweep on $\delta$ reveals a phase transition near $\delta = 0$: accuracy is low in subcritical regime, while supercritical initialization sustains high performance for bifurcating activations. It should be noted that the transition is not as sharp as in the toy model, because random matrix theory establishes deterministic results in the limit of large embedding dimension.

Interestingly, ReLU in the supercritical regime maintains a non-zero $E_D^0$, but poor accuracy. This is consistent with our theory: ReLU lacks the stabilizing cubic term ($\gamma = \phi'''(0) < 0$) required to align with graph eigenmodes. In contrast, bifurcating activations provide both the instability to escape the homogeneous state *and* stabilization towards expressive structured patterns.

*Remark* 6.1. This shows that breaking contractivity alone is insufficient to mitigate oversmoothing: higher-order nonlinear terms are essential for stabilizing informative patterns.

### 6.2. Node Classification

Table 1 presents results across benchmarks of varying size and homophily. Comparisons with state-of-the-art models illustrate that our approach yields competitive results without sophisticated architectures, across all levels of homophily. Details of the experiments in Appendix B.

We emphasize that these benchmarks evaluate shallow networks where oversmoothing is not the limiting factor. Our goal here is not to achieve state-of-the-art performance, but to demonstrate that activations do not compromise expres-

*Table 1.* **Node classification accuracy (%) on benchmark datasets.** Results are reported as mean ± standard deviation across multiple runs. Datasets are ordered by increasing homophily levels, where higher values indicate more homophilic graphs. Gold, silver, and bronze indicate 1st, 2nd, and 3rd best results. Results for baseline methods on ogbn-ArXiv are taken from the corresponding papers.

| Method | Texas | Wisconsin | Squirrel | Chameleon | Cornell | Computer | CiteSeer | PubMed | Cora | CoauthorCS | Photo | ogbn-ArXiv |
|---|---|---|---|---|---|---|---|---|---|---|---|---|
| *Homophily* | *0.11* | *0.21* | *0.22* | *0.23* | *0.30* | *0.78* | *0.74* | *0.80* | *0.81* | *0.81* | *0.83* | 0.83 |
| GCN | 55.14 ± 5.16 | 51.76 ± 3.06 | 53.43 ± 2.01 | 64.82 ± 2.24 | 60.54 ± 5.30 | 82.6 ± 2.4 | 76.50 ± 1.36 | 88.42 ± 0.50 | 86.98 ± 1.27 | 91.1 ± 0.5 | 91.2 ± 1.2 | 72.17 ± 0.33 |
| GCN + PairNorm | 60.27 ± 4.34 | 48.43 ± 6.14 | 50.44 ± 2.04 | 62.74 ± 2.82 | 58.92 ± 3.15 | 73.59 ± 1.47 | – | 87.53 ± 0.44 | 85.79 ± 1.01 | – | – | – |
| GraphSAGE | 82.43 ± 6.14 | 81.18 ± 5.56 | 41.61 ± 0.74 | 58.73 ± 1.68 | 75.95 ± 5.01 | 82.4 ± 1.8 | 76.04 ± 1.30 | 88.45 ± 0.50 | 86.90 ± 1.04 | 91.3 ± 2.8 | 91.4 ± 1.3 | – |
| GAT | 52.16 ± 6.63 | 49.41 ± 4.09 | 40.72 ± 1.55 | 60.26 ± 2.50 | 61.89 ± 5.05 | 78.0 ± 19.0 | 76.55 ± 1.23 | 87.30 ± 1.10 | 86.33 ± 0.48 | 90.5 ± 0.6 | 85.7 ± 20.3 | 73.65 ± 0.11 |
| CGNN | 71.35 ± 4.05 | 74.31 ± 7.26 | 29.24 ± 1.09 | 46.89 ± 1.66 | 66.22 ± 7.69 | 80.29 ± 2.0 | 76.91 ± 1.81 | 87.70 ± 0.49 | 87.10 ± 1.35 | 92.3 ± 0.2 | 91.39 ± 1.5 | 57.80 ± 2.5 |
| GRAND | 75.68 ± 7.25 | 79.41 ± 3.64 | 40.05 ± 1.50 | 54.67 ± 2.54 | 74.59 ± 4.04 | 69.12 ± 0.41 | 76.46 ± 1.77 | 89.02 ± 0.51 | 87.36 ± 0.96 | 92.89 ± 0.30 | 84.64 ± 0.25 | 72.23 ± 0.20 |
| GREAD | 84.59 ± 4.53 | 85.29 ± 4.49 | 58.62 ± 1.08 | 70.00 ± 1.70 | 73.78 ± 4.53 | 69.89 ± 0.45 | 77.09 ± 1.78 | 90.01 ± 0.42 | 88.16 ± 0.81 | 92.93 ± 0.13 | 85.81 ± 0.63 | – |
| SGOS-Expn | 88.65 ± 2.64 | 88.82 ± 3.62 | 50.83 ± 1.69 | 69.74 ± 1.26 | 76.48 ± 2.71 | 70.35 ± 0.24 | 77.48 ± 1.26 | 90.08 ± 0.49 | 88.31 ± 0.85 | 93.71 ± 0.23 | 85.49 ± 0.98 | – |
| ReLU-Poly(A)-sh | 92.79 ± 0.1 | 86.27 ± 0.1 | 53.57 ± 0.82 | 69.63 ± 0.35 | 67.57 ± 0.2 | 91.35 ± 0.55 | 77.56 ± 0.66 | 88.44 ± 0.07 | 89.33 ± 0.38 | 93.43 ± 0.09 | 94.27 ± 0.11 | 72.95 ± 0.15 |
| ReLU-Poly(A)-pl | 90.09 ± 0.1 | 84.31 ± 0.0 | 53.67 ± 0.46 | 69.89 ± 0.49 | 76.58 ± 0.1 | 91.43 ± 0.33 | 77.68 ± 0.74 | 88.29 ± 0.18 | 89.33 ± 0.20 | 93.47 ± 0.09 | 94.24 ± 0.12 | 73.17 ± 0.22 |
| Sine-Poly(A)-sh | 85.59 ± 0.1 | 85.62 ± 0.0 | 60.81 ± 0.56 | 70.81 ± 0.24 | 76.58 ± 0.1 | 91.56 ± 0.32 | 78.18 ± 1.01 | 89.71 ± 0.09 | 89.78 ± 0.20 | 95.84 ± 0.10 | 95.69 ± 0.17 | 73.29 ± 0.8 |
| Sine-Poly(A)-pl | 93.69 ± 0.2 | 80.39 ± 0.0 | 61.79 ± 0.40 | 70.85 ± 0.92 | 76.87 ± 0.1 | 91.94 ± 0.07 | 78.15 ± 0.75 | 88.67 ± 0.27 | 89.89 ± 0.50 | 95.42 ± 0.07 | 95.55 ± 0.21 | 72.77 ± 0.34 |

| Method | Mean Rank |
|---|---|
| Sine-Poly(A)-sh | 3.00 ± 2.13 |
| Sine-Poly(A)-pl | 3.18 ± 3.31 |
| GELU-Poly(A)-pl | 4.82 ± 1.78 |
| GELU-Poly(A)-sh | 5.32 ± 2.56 |
| ReLU-Poly(A)-pl | 6.64 ± 2.41 |
| ReLU-Poly(A)-sh | 6.91 ± 3.56 |
| Swish-Poly(A)-pl | 7.36 ± 2.26 |
| Swish-Poly(A)-sh | 7.59 ± 2.35 |
| Sine GCN | 10.73 ± 6.08 |

*Table 2.* **Ablation by average rank (± std) across benchmark datasets.** Methods are ranked by accuracy on each dataset (1 = best), and the average rank is computed across all datasets except large-scale OGBN-arXiv. Lower is better.

sivity in standard settings. The key validation of our theory lies in the deep regime (Figure 5), where sine, tanh and Fischer-tanh activated networks maintain performance at 64 layers while ReLU-based counterparts collapse.

### 6.3. Ablation

We ablate the activation and the polynomial filtering on the node-classification task to single out their performance contribution. Namely, beyond the ReLU-Poly and Sine-Poly activation and Sine-GCN without polynomial filtering, we also try alternative GeLU-Poly and Swish-Poly, where we replace the base activation functions with Gaussian Error Linear Units (GELU) (Hendrycks & Gimpel, 2016) and Swish (Ramachandran et al., 2017). The summary results are shown in Table 2. Notably, our proposed Sine-Poly(A) is consistently among the best choices across the benchmarks, and the polynomial filtering is key to this choice, as only replacing the activation (Sine-GCN) yields similar results as the vanilla GCN. Moreover, our results hold for all alternative activation functions, with competitive results even with GELU or Swish (those alternatives are ranked just after the Sine-Poly proposal). This ablation empirically validates the robust combination of bifurcating activations

with polynomial filter models to gain expressivity on real datasets while mitigating oversmoothing. The full ablation comparison is available in Appendix Table 6.

## 7. Conclusion

This work reframes oversmoothing as a dynamical stability problem. We demonstrated that more general activations can instantiate bifurcating systems: specifically, replacing standard ReLU with functions with stabilizing cubic nonlinearities enables deep GNNs to stabilize non-homogeneous patterns rather than collapsing to triviality. Our analysis, although local, yields precise amplitude scaling laws and a closed-form initialization strategy that positions networks at criticality, creating a strong topological prior. Finally, we show that these activation functions do not come at the expense of poor expressivity in standard tasks. In fact, combined with polynomial mode filtering, these architectures yield robust and competitive results on a node-classification benchmark.

A current limitation is our reliance on fixed graph topologies: while we showed that filters can *select* specific eigenmodes as priors, the network is restricted to the static "menu" of the input graph. A promising future direction is to extend this bifurcation framework to learnable graph architectures such as attention-based mechanisms (Veličković et al., 2018). By coupling feature dynamics with an evolving topology, models could go beyond selecting static priors to actively **sculpting the eigenmodes themselves**.

## Impact Statement

This paper presents work whose goal is to advance the field of machine learning. There are many potential societal consequences of our work, none of which we feel must be specifically highlighted here.

## Acknowledgments

Parts of this work were supported by the ERC Consolidator Grant 101087347 (VEGA), as well as gifts from Ansys Inc., and Adobe Research.

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

# A. Theoretical Results and Proofs

This appendix gathers the analytical background and detailed proofs supporting the theoretical results presented in Sections 3,4 and 5 of the main text. It is organized to guide the reader through the minimal tools and reasoning underlying the bifurcation analysis of Graph Neural Networks (GNNs).

A.1 shows how the linear mapping of Eq.2 can be translated into a dynamical system problem, with an effective potential which dictates its stability. A.2 provides an intuitive potential landscape interpretation of the GNN nonlinear steady states and their stability transitions. A.3 provides an analogy of the problem with phase transition theory.

The following subsections provide full proofs of the main results: A.4 proves Theorem 3.2 (spontaneous symmetry breaking in a single-layer GNN), A.5 demonstrates the Dirichlet energy bifurcation of Corollary 3.4 and A.6 for the effective potential energy formulation of Corollary 3.5).

The generalization to the multi-feature, random-weight extension in Theorem 5.1 is proven in A.9. Section A.10 summarizes the classical results of random matrix theory that determine the spectral scaling and critical variance conditions for random feature weights in Theorem 5.1 and Corollary 5.2. Together, these results establish a unified bifurcation-theoretic understanding of how graph structure, nonlinearities, and initialization interact to determine the emergence of structured patterns in deep GNNs.

## A.1. Stability and effective potential

We view the iterative layer update as a discrete gradient descent step on the effective potential $V(x)$. From Equation 3, we defined the potential such that $-V'(x) = \phi(wx) - x$.

This means the layer update fundamentally pushes the state $x$ in the direction of the negative gradient of $V$:

$$x^{(l+1)} - x^{(l)} = -V'(x^{(l)})$$

To rigorously illustrate why local minima are stable and maxima are unstable, it is helpful to look at the continuous-time analog of this system:

$$\frac{dx}{dt} = -V'(x)$$

Let $x^*$ be a fixed point, meaning $V'(x^*) = 0$. If we introduce a small perturbation $\delta x(t) = x(t) - x^*$ and take the second-order Taylor expansion of the potential around $x^*$, we get:

$$V(x) \approx V(x^*) + V'(x^*)\delta x + \frac{1}{2}V''(x^*)(\delta x)^2$$

Since $V'(x^*) = 0$, the gradient of the potential near the fixed point is linearly approximated by $V'(x) \approx V''(x^*)\delta x$. Substituting this into our continuous-time dynamics yields the linear ordinary differential equation:

$$\frac{dx}{dt} = \frac{d(\delta x)}{dt} = -V''(x^*)\delta x$$

The solution to this ODE governs the behavior of the perturbation over time:

$$\delta x(t) = \delta x(0)e^{-V''(x^*)t}$$

This explicitly demonstrates the stability criterion based on the curvature $V''(x^*)$:

- **Local Minimum ($V''(x^*) > 0$):** The exponent $-V''(x^*)$ is negative, causing the perturbation to **decay exponentially** to zero. The system returns to $x^*$, making it a **stable** fixed point.

- **Local Maximum ($V''(x^*) < 0$):** The exponent $-V''(x^*)$ is positive, causing the perturbation to **diverge exponentially**. The system is pushed away from $x^*$, making it an **unstable** fixed point.

## A.2. Stable states of the effective potential

The effective potential offers an intuitive and compact way to describe the appearance and stability of new steady states in systems depending on a control parameter (here $w$). The idea is to represent the effective dynamics of a scalar order parameter $a$ by an energy function (or Landau energy) $V(a; w)$ whose minima correspond to the stable equilibria.

**Basic construction.** When the reduced fixed-point condition can be written as

$$0 = A(w)a + B(w)a^3 + O(a^5),$$

with $B(w) > 0$, one defines the Landau energy

$$V(a; w) = \frac{A(w)}{2} a^2 + \frac{B(w)}{4} a^4 + O(a^6), \qquad -\frac{\partial V}{\partial a} = 0$$

as the equilibrium condition. The shape of $V$ changes qualitatively when $A(w)$ changes sign.

**Interpretation.**

- $A(w) > 0$ — single minimum at $a = 0$ (symmetric or homogeneous phase);

- $A(w) < 0$ — two symmetric minima at $a = \pm a_\star$ (symmetry-broken phase), with $a_\star = \sqrt{-A(w)/B(w)} + O(|A|^{3/2})$.

The coefficient $A(w)$ acts as the "distance to criticality," and $a$ serves as the *order parameter*. In our analysis, $A(w) = \mu = \alpha w \lambda_k - 1$ and $B(w) = \frac{-\gamma(w\lambda_k)^3 \kappa_k}{6}$, so the transition occurs at $w\lambda_k = 1$.

**Intuitive role in this paper.** The Landau energy condenses the nonlinear dynamics of the GNN layer into a scalar potential:

$$V(a; w) = -\frac{\alpha w \lambda_k - 1}{2} a^2 + \frac{\gamma(w\lambda_k)^3 \kappa_k}{24} a^4.$$

Its curvature change at $\alpha w \lambda_k = 1$ signals the loss of stability of the homogeneous state and the emergence of two symmetry-related minima. This "energy landscape" picture is directly analogous to a continuous phase transition or spontaneous magnetization in physics, and provides an intuitive way to visualize the bifurcation results proved in Theorem 3.2 and Corollary 3.3.

## A.3. Analogy with phase transitions

This linear onset of energy is directly analogous to a **second-order (continuous) phase transition** in statistical physics. In the Ising or ferromagnetic Landau picture, the magnetization $M$ satisfies

$$M \propto \sqrt{T_c - T},$$

below the critical temperature $T_c$, while the free energy increases quadratically with $(T_c - T)$. Here, the **amplitude** $a_\star$ plays the role of the spontaneous magnetization and the **Dirichlet energy** $E_D \propto a_\star^2$ behaves like the square of that order parameter, growing linearly with the distance $\mu = \alpha w \lambda_k - 1$ from criticality.

Consequently, the trivial homogeneous state corresponds to the **disordered phase**, while the bifurcated non-homogeneous states $x_\pm^\star$ correspond to two **symmetry-broken phases** (positive and negative magnetization). This analogy naturally motivates the **Landau energy functional** introduced in the next result (Corollary 3.5), which provides a potential-energy landscape whose minima reproduce this bifurcation behavior.

## A.4. Proof of Theorem 3.2

*Proof.* We prove the result in five steps, following a Lyapunov–Schmidt reduction.

**Step 1 (Mode decomposition).** Let $A$ be the normalized graph adjacency matrix with eigenpairs $(\lambda_r, u_r)$, where

$$0 = \lambda_0 < \lambda_1 \le \cdots \le \lambda_{k-1} < \lambda_k$$

and assume $\lambda_k$ is simple. Fix $w > 0$ and consider fixed points of

$$x = \phi(wAx).$$

Where we assume the activation $\phi$ is *odd* and $\phi'(0) = \alpha > 0, \phi'''(0) = -\gamma < 0$. Let us decompose $x$ along $u_k$ and its orthogonal complement:

$$x = au_k + y, \qquad a \in \mathbb{R}, \quad y \in \mathcal{X}_\perp := \{u_k\}^\perp.$$

Let $P = u_k u_k^\top$ and $Q = I - P$. Projecting the fixed-point equation onto $\mathrm{span}\{u_k\}$ and $\mathcal{X}_\perp$ yields

$$\langle u_k, \phi(wA(au_k + y))\rangle = a, \tag{11}$$

$$y = Q\phi(wA(au_k + y)). \tag{12}$$

We will solve (12) for $y = y(a, w)$ and insert it into (11).

**Step 2 (Transverse equation).** Since $y \perp u_k$, the relevant linear operator on $\mathcal{X}_\perp$ is $QAQ^T$, whose spectrum is $\{\lambda_r : r \ne k\}$. Choose $w$ in a one-sided neighborhood of

$$w_k := \frac{1}{\lambda_k}$$

such that

$$w\lambda_r < 1 \quad \text{for all } r \ne k. \tag{13}$$

This is possible precisely because we assumed $\lambda_k$ is the largest Laplacian eigenvalue: then $w \approx 1/\lambda_k$ automatically satisfies $w\lambda_r \le w\lambda_{k-1} < 1$ for all $r \ne k$. Under (13), the operator $(I - wQAQ^T)$ is invertible on $\mathcal{X}_\perp$ and

$$\|(I - wQAQ^T)^{-1}\| \le \frac{1}{1 - wk_{r\ne k}\lambda_r}.$$

Expand the activation entrywise:

$$\phi(z) = \alpha z + \frac{\gamma}{6}z^3 + R_5(z), \qquad |R_5(z)_i| \le C|z_i|^5,$$

since $\phi$ is odd and use $Lu_k = \lambda_k u_k$ to rewrite (12) as

$$y = Q\Big(\alpha wA(au_k + y) - \frac{\gamma w^3}{6}(A(au_k + y))^{\odot 3} + R_5(wA(au_k + y))\Big)$$

$$= \alpha wQAQ^T y - \frac{\gamma w^3}{6}Q((\lambda_k au_k + Ay)^{\odot 3}) + QR_5(wA(au_k + y)).$$

Bring the linear term to the left:

$$(I - \alpha wQAQ^T)y = -\frac{\gamma w^3}{6}Q((\lambda_k au_k + Ay)^{\odot 3}) + QR_5(wA(au_k + y)).$$

Apply $(I - wQAQ^T)^{-1}$ to obtain

$$y = \mathcal{T}(y; a, w),$$

where $\mathcal{T}$ collects the cubic and higher-order terms. Using $\|y\|$ as the norm on $\mathcal{X}_\perp$, the bounds on $R_5$, and that $y$ will turn out to be of order $a^3$, one checks that for $|a|$ small enough, $\mathcal{T}$ maps a ball $\{y : \|y\| \le C|a|^3\}$ into itself and is a contraction. Hence, for $|a|$ small and $w$ close to $w_k$, (12) admits a unique solution

$$y = y(a, w) \in \mathcal{X}_\perp$$

such that

$$\|y(a, w)\| \le C|a|^3.$$

**Step 3 (Reduced scalar equation).** Define

$$G(a, w) := \langle u_k, \phi(wA(au_k + y(a, w))) \rangle - a.$$

We now expand $G$ in $a$ near $(a, w) = (0, w_k)$. Using again the Taylor expansion of $\phi$ and $Au_k = \lambda_k u_k$,

$$\langle u_k, \phi(wA(au_k + y)) \rangle = \langle u_k, \alpha wA(au_k + y) \rangle - \frac{\gamma}{6}\langle u_k, (wA(au_k + y))^{\odot 3} \rangle + O(\|a\|^5)$$

$$= \alpha w \lambda_k a - \frac{\gamma w^3}{6} \sum_{i=1}^{n} (\lambda_k a u_{k,i} + (Ay)_i)^3 u_{k,i} + O(a^5).$$

Since $y = O(a^3)$, all cross-terms involving $Ay$ are $O(a^5)$ and can be absorbed in the remainder. The leading cubic term is

$$\sum_{i=1}^{n} (\lambda_k a u_{k,i})^3 u_{k,i} = \lambda_k^3 a^3 \sum_{i=1}^{n} u_{k,i}^4 = \lambda_k^3 a^3 \kappa_k,$$

where $\kappa_k := \sum_{i=1}^{n} u_{k,i}^4 > 0$. Therefore,

$$G(a, w) = (\alpha w \lambda_k - 1)a - \frac{\gamma w^3 \lambda_k^3}{6}\kappa_k a^3 + R(a, w), \qquad |R(a, w)| \leq C|a|^5.$$

**Step 4 (Nontrivial solutions and amplitude).** Factor

$$G(a, w) = a\, H(a, w), \qquad H(a, w) = (\alpha w \lambda_k - 1) - \frac{\gamma w^3 \lambda_k^3}{6}\kappa_k a^2 + O(a^4).$$

Let $\mu := \alpha w \lambda_k - 1$. For $w > w_k$ we have $\mu > 0$ and $H(0, w) = \mu > 0$. For $a$ large enough (but still small in absolute terms), the negative quadratic term dominates, so $H(a, w) < 0$. By continuity, there exists $a_\star^+(w) > 0$ such that $H(a_\star^+(w), w) = 0$, hence $G(a_\star^+(w), w) = 0$. By oddness in $a$, there is a symmetric solution $a_\star^-(w) = -a_\star^+(w)$. Solving the leading-order balance

$$\mu - \frac{\gamma w^3 \lambda_k^3}{6}\kappa_k a^2 = 0$$

gives

$$a_\star(w) = \sqrt{\frac{6\mu}{\gamma w^3 \lambda_k^3 \kappa_k}} + O(\mu^{3/2}).$$

Thus the nontrivial fixed points are

$$x_\pm^\star(w) = \pm a_\star(w) u_k + O(a_\star(w)^3),$$

which is the claimed square-root bifurcation.

**Step 5 (Local stability).** Let $x^\star$ be either branch. The Jacobian of the map $x \mapsto \phi(wAx)$ at $x^\star$ is

$$J^\star = \mathrm{diag}(\phi'(wAx^\star))\, wA.$$

Because $x^\star = \pm a_\star u_k + O(a_\star^3)$, we have

$$wAx^\star = w\lambda_k a_\star u_k + O(a_\star^3).$$

Expanding $\phi'(z) = \alpha - \frac{\gamma}{2}z^2 + O(z^4)$ entrywise and projecting onto $u_k$ gives the multiplier in the critical direction:

$$m_k = \langle u_k, J^\star u_k \rangle$$

$$= w\lambda_k \sum_{i=1}^{n} \left( \alpha - \frac{\gamma}{2}(w\lambda_k a_\star u_{k,i})^2 + O(a_\star^4) \right) u_{k,i}^2$$

$$= \alpha w \lambda_k - \frac{\gamma w^3 \lambda_k^3}{2}\kappa_k a_\star^2 + O(a_\star^4).$$

Substitute $a_\star^2 = \frac{6\mu}{\gamma w^3 \lambda_k^3 \kappa_k} + O(\mu^2)$:

$$m_k = (1 + \mu) - 3\mu + O(\mu^2) = 1 - 2\mu + O(\mu^2),$$

so for $0 < \mu \ll 1$ we have $|m_k| < 1$. For the transverse modes $r \neq k$, the multipliers are

$$m_r = w\lambda_r + O(a_\star^2),$$

and by (13) these satisfy $|m_r| < 1$ for $w$ close enough to $w_k$. Hence the spectral radius of $J^\star$ is $< 1$, and the two nontrivial fixed points are locally exponentially stable. $\square$

## A.5. Proof of Corollary 3.4 (Dirichlet energy bifurcation)

*Proof.* Let $x_\pm^\star(w)$ denote the two nontrivial fixed points of Theorem 3.2 for $w > w_k = 1/\lambda_k$. Recall that

$$x_\pm^\star(w) = \pm a_\star(w)\, u_k + O(a_\star(w)^3), \qquad a_\star(w) = \sqrt{\frac{6\mu}{\gamma w^3 \lambda_k^3 \kappa_k}} + O(\mu^{3/2}),$$

where $\mu := \alpha w \lambda_k - 1$ and $\kappa_k = \sum_i u_{k,i}^4 > 0$.

**Step 1 (Energy of the homogeneous state).** For the trivial fixed point $x_0^\star = 0$,

$$E_D(x_0^\star) = (x_0^\star)^\top L x_0^\star = 0.$$

Thus the Dirichlet energy vanishes for all $w \leq w_k$, corresponding to the completely homogeneous (oversmoothed) regime.

**Step 2 (Energy of the bifurcated branch).** Substituting $x_\pm^\star(w)$ into the quadratic form $E_D(x) = x^\top L x$ gives

$$\begin{aligned}
E_D(x_\pm^\star) &= (\pm a_\star u_k + O(a_\star^3))^\top L(\pm a_\star u_k + O(a_\star^3)) \\
&= a_\star^2\, u_k^\top L u_k + O(a_\star^4) \\
&= a_\star^2\, u_k^\top (I - A) u_k + O(a_\star^4) = (1 - \lambda_k) a_\star^2 + O(a_\star^4).
\end{aligned}$$

Using $a_\star^2 = \frac{6\mu}{\gamma w^3 \lambda_k^3 \kappa_k} + O(\mu^2)$, we obtain

$$E_D(x_\pm^\star) = \frac{6(1 - \lambda_k)\mu}{\gamma(w\lambda_k)^3 \kappa_k} + O(\mu^2).$$

**Step 3 (Scaling law).** Because $\lambda_k > 0$, $\kappa_k > 0$, and $\mu > 0$ for $w > w_k$, the Dirichlet energy increases linearly with $\mu$:

$$E_D(x_\pm^\star) = \begin{cases} 0, & w \leq w_k, \\ C_k(w\lambda_k - 1) + O((w\lambda_k - 1)^2), & w > w_k, \end{cases}$$

where $C_k = \frac{6(1-\lambda_k)}{\gamma(w\lambda_k)^3 \kappa_k} > 0$. Hence $E_D$ serves as an order parameter for the loss of homogeneity.

$\square$

## A.6. Effective potential (Corollary 3.5)

*Idea and construction.* Near the critical coupling $w_k = 1/(\alpha\lambda_k)$, the amplitude $a$ of the dominant eigenmode $u_k$ satisfies, up to cubic order,

$$0 = (\alpha w \lambda_k - 1)a - \frac{\gamma w^3 \lambda_k^3}{6} \kappa_k\, a^3 + O(a^5), \qquad \kappa_k = \sum_i u_{k,i}^4 > 0.$$

Rather than viewing this as a stand-alone algebraic relation, we can interpret it as the stationarity condition of a scalar *energy function* $V(a; w)$ in the sense that

$$-\frac{\partial V}{\partial a}(a; w) = (\alpha w \lambda_k - 1)a - \frac{\gamma w^3 \lambda_k^3}{6} \kappa_k\, a^3.$$

Integrating in $a$ yields, up to fourth order,

$$V(a; w) = \frac{1 - \alpha w \lambda_k}{2} a^2 + \frac{\gamma(w \lambda_k)^3}{24} \kappa_k a^4 + O(a^6). \tag{14}$$

The fixed points of the layer therefore correspond to the *minima* of this energy landscape.

**Interpretation.** The quadratic coefficient in (14) changes sign at $\alpha w \lambda_k = 1$:

- For $\alpha w \lambda_k < 1$, it is positive and $V(a; w)$ has a single minimum at $a = 0$, representing the homogeneous (oversmoothed) state.

- For $\alpha w \lambda_k > 1$, it becomes negative while the quartic term remains positive, turning $V$ into a symmetric double well with two minima at

$$a_\star(w) = \sqrt{\frac{6(\alpha w \lambda_k - 1)}{\gamma(w \lambda_k)^3 \kappa_k}} + O((\alpha w \lambda_k - 1)^{3/2}),$$

corresponding to the two non-homogeneous stable states $x_\pm^\star = \pm a_\star u_k$.

**Analogy with phase transitions.** The energy (14) has the same form as the Landau free energy in mean-field models of ferromagnetism. Here, the amplitude $a$ plays the role of the magnetization: before the critical point ($w \lambda_k < 1$) the energy has one minimum at $a = 0$ (disordered phase), while after the transition ($w \lambda_k > 1$) two symmetric minima appear (spontaneous symmetry breaking). The system "chooses" one minimum, breaking the $a \mapsto -a$ symmetry, just as a magnet acquires a spontaneous orientation below its Curie temperature.

**Summary.** Equation (14) thus provides a compact and intuitive potential whose minima coincide with the stable fixed points of the GNN layer. Using $-\partial V / \partial a = 0$ clarifies that these states correspond to *energy minima*, making the stability and the phase-transition analogy transparent to readers from both physics and machine-learning backgrounds. $\qquad \square$

### A.7. Neural Tangent Kernel Framework and Proof of Corollary 4.1

**Hitchhiker's guide to NTK** :

This section serves as a "Hitchhiker's Guide" to the Neural Tangent Kernel (NTK), providing the minimal theoretical machinery needed to understand how bifurcation creates a topological learning prior. We first review the general framework introduced by Jacot et al. and then apply it to prove Corollary 4.1.

FROM GRADIENT DESCENT TO KERNEL EVOLUTION

To see how standard gradient descent becomes a kernel process, we move from parameter space to function space using the chain rule. Consider a network function $f_\theta$ trained to minimize a loss $\mathcal{L}(f)$ using gradient descent on parameters $\theta$. The parameter update follows the negative gradient:

$$\frac{d\theta}{dt} = -\eta \nabla_\theta \mathcal{L} = -\eta (\nabla_f \mathcal{L}) \nabla_\theta f, \tag{15}$$

where $\eta$ is the learning rate. We are interested in how the function output $f$ itself evolves. By the chain rule:

$$\frac{df}{dt} = (\nabla_\theta f)^T \frac{d\theta}{dt}. \tag{16}$$

Substituting the parameter update equation into this function update yields:

$$\frac{df}{dt} = (\nabla_\theta f)^T \left(-\eta (\nabla_f \mathcal{L}) \nabla_\theta f\right) = -\eta \underbrace{(\nabla_\theta f)^T (\nabla_\theta f)}_{K} \nabla_f \mathcal{L}. \tag{17}$$

Here, $K(x, x') = \langle \nabla_\theta f(x), \nabla_\theta f(x') \rangle$ is the **Neural Tangent Kernel (NTK)**. It acts as a similarity matrix that dictates how an update at data point $x$ affects the prediction at $x'$.

RATIONALIZING LEARNING SPEEDS VIA MODE DECOMPOSITION

Why do some patterns learn faster than others? This is determined by the spectrum (eigenvalues) of the kernel $K$. Consider a standard Least-Squares regression task where the loss is $\mathcal{L} = \frac{1}{2}\|f - y\|^2$. The gradient with respect to function outputs is simply the error residual: $\nabla_f \mathcal{L} = (f - y)$.

The dynamics equation becomes a linear differential equation:

$$\frac{d(f - y)}{dt} = -\eta K(f - y). \tag{18}$$

We can solve this by decomposing the error into the eigenbasis of the kernel $K$. Let $(v_r, \lambda_r^{(K)})$ be the eigenvector-eigenvalue pairs of $K$, such that $Kv_r = \lambda_r^{(K)} v_r$. If we write the residual vector as a sum of these modes, $(f - y)(t) = \sum_r c_r(t) v_r$, the evolution decouples:

$$\frac{dc_r}{dt} = -\eta \lambda_r^{(K)} c_r \quad \implies \quad c_r(t) = c_r(0) e^{-\eta \lambda_r^{(K)} t}. \tag{19}$$

This reveals a fundamental spectral bias: the convergence rate for the $r$-th mode is proportional to its kernel eigenvalue $\lambda_r^{(K)}$. Modes with large $\lambda_r^{(K)}$ are learned exponentially fast, while modes with small $\lambda_r^{(K)}$ are learned very slowly.

**Proof of Corollary**

*Proof.* We consider the two regimes separately.

**Supercritical regime** $(w > w_k)$. Let $\mu := \alpha w \lambda_k - 1$. By Theorem 3.2, for $0 < \mu \ll 1$ the dynamics admit exactly two stable nonzero fixed points

$$x_\pm^\star(w) = \pm a^\star(w) u_k + O\big((a^\star(w))^3\big), \tag{20}$$

where

$$a^\star(w) = \sqrt{\frac{6\mu}{\gamma(w\lambda_k)^3 \kappa_k}} + O(\mu^{3/2}), \qquad \kappa_k := \sum_i u_{k,i}^4. \tag{21}$$

Differentiating the fixed-point expansion with respect to $w$ yields

$$\partial_w x_\pm^\star(w) = \pm a^{\star\prime}(w) u_k + O(1), \tag{22}$$

since $(a^\star(w))^3 = O(\mu^{3/2})$ and $\partial_w \mu = \alpha \lambda_k$.

From the leading-order expression for $a^\star(w)$ we obtain

$$a^{\star\prime}(w) = \Theta(\mu^{-1/2}) + O(1), \tag{23}$$

and therefore

$$\partial_w x_\pm^\star(w) = \frac{c}{\sqrt{\mu}} u_k + O(1) \tag{24}$$

for some constant $c > 0$.

Forming the NTK,

$$K_w(i, j) = \partial_w x^\star(i) \, \partial_w x^\star(j) = \frac{c^2}{\mu} u_k(i) u_k(j) + O(1), \tag{25}$$

which proves the first claim.

**Subcritical regime** $(w < w_k)$. For $w < w_k$, the homogeneous fixed point $x^\star = 0$ is locally stable (Theorem 3.2). Using the activation expansion $\phi(z) = \alpha z + O(z^3)$, the linearized update map is

$$x \mapsto \alpha w A x, \tag{26}$$

whose spectral decomposition is given by the eigenpairs $(\lambda_r, u_r)$ of $A$.

Since $|\alpha w \lambda_r| < 1$ for all $r$, sensitivities remain bounded and are governed by the resolvent $(I - \alpha w A)^{-1}$. The NTK therefore scales as the square of this resolvent:

$$K_w \;\propto\; (I - \alpha w A)^{-2}. \tag{27}$$

Diagonalizing $A$ yields

$$K_w(i,j) \;\propto\; \sum_r \frac{1}{(1 - \alpha w \lambda_r)^2} u_r(i) u_r(j), \tag{28}$$

which completes the proof. $\qquad\square$

### A.8. Full Theorem

**Theorem A.1** (Bifurcation on realistic GNNs). *Consider the discrete layer update*

$$X^{(\ell+1)} = \phi\big(A\,X^{(\ell)}\,W\big), \tag{29}$$

*where $A \in \mathbb{R}^{n \times n}$ is the normalized graph adjacency matrix, $W \in \mathbb{R}^{d \times d}$ is a random feature-weight matrix. The entries of $W$ are i.i.d. with $\mathbb{E}[W_{ij}] = 0$ and $\mathrm{Var}(W_{ij}) = v$, and $\phi \in C^3(\mathbb{R})$ an odd function which acts entry-wise with $\phi'(0) = \alpha > 0$ and $\phi'''(0) = -\gamma < 0$. Define the vectorized variable $x = \mathrm{vec}(X) \in \mathbb{R}^{nd}$ and the matrix $\tilde{A} = W^\top \otimes A$, so that the update can be written equivalently as $x^{(\ell+1)} = \phi(\tilde{A}\,x^{(\ell)})$,*

*Let $\lambda_k$ be the largest eigenvalue of $A$, and let $\rho(W)$ denote the spectral radius of $W$. Then:*

1. *The homogeneous fixed point $x = 0$ is locally stable if and only if all the eigenvalues of $\tilde{A}$ lie in the open unit disk and loses stability when*

   $$\alpha \rho(W)\,\lambda_k = 1. \tag{30}$$

   *At this critical value, the system undergoes its first (supercritical) pitchfork bifurcation.*

2. *For $\alpha \rho(W)\,\lambda_k > 1$ but close to 1, two stable, nontrivial symmetry-related equilibria emerge. Each corresponds to a rank-one activation pattern aligned with a graph mode $u_k$ and a feature mode $v_j$:*

   $$X^\star_{\pm,k,j} = \pm\, a^\star_{k,j}\, u_k v_j^\top + O\big((a^\star_{k,j})^3\big),$$

   *where $u_k$ (resp. $v_j$) is the eigenvector associated to the leading eigenvalue $\lambda_k$ of $A$ (resp. $\sigma_j$ of $W$). The amplitude of the pattern satisfies*

   $$a^\star_{k,j} = \sqrt{\frac{6\,[\,\alpha\lambda_k|\sigma_j| - 1\,]}{\gamma(\lambda_k\sigma_j)^3\,\kappa_k\,\xi_j}} + O\big((\alpha\lambda_k|\sigma_j| - 1)^{3/2}\big),$$

   *with $\kappa_k = \sum_i u_{k,i}^4$ and $\xi_j = \sum_i v_{j,i}^4$.*

Theorem 5.1 is a non-trivial extension of our 1D result, proving the bifurcation mechanism is not merely an artifact of a simplified system. The bifurcation condition in (30) reflects an elegant interaction between the dominant spatial frequency of the graph ($\lambda_k$) and the dominant feature-mixing factor ($\rho(W)$). When their product exceeds 1, the homogeneous fixed point becomes unstable.

Crucially, emergent stable states are no longer simple vectors; they are *rank-one tensors* of the form $u_k v_j^\top$. This has a clear and powerful interpretation: the emerging stable pattern is a separable state composed of a specific *graph eigenmode* ($u_k$) and a corresponding *feature co-activation pattern* ($v_j$). In other words, the GNN's dynamics naturally stabilize a pattern that is simultaneously aligned with the graph structure and the learned feature space.

### A.9. Proof of Theorem 5.1

We prove that the matrix-valued update

$$X^{(\ell+1)} = \phi\big(AX^{(\ell)}W\big)$$

undergoes a first bifurcation when $\alpha\rho(W)\lambda_k(A) = 1$, and that the emerging equilibria are rank-one patterns $u_k v_j^\top$ with amplitude given in the statement.

**Step 1: Vectorized form and linearization.** Let $x = \text{vec}(X) \in \mathbb{R}^{nd}$ and recall the identity

$$\text{vec}(AXW) = (W^\top \otimes L)\,\text{vec}(X).$$

Define

$$\tilde{A} := W^\top \otimes A \in \mathbb{R}^{nd \times nd}.$$

Then the layer update can be written entrywise as

$$x^{(\ell+1)} = \phi(\tilde{A}x^{(\ell)}),$$

with $\phi$ acting componentwise. To study the stability of $x = 0$, expand $\phi(z) = \alpha z + O(z^3)$ to get the linearized map

$$x^{(\ell+1)} = \alpha\tilde{A}x^{(\ell)} + O(\|x^{(\ell)}\|^3).$$

Therefore $x = 0$ is locally stable iff the spectral radius of $\tilde{A}$ is $< 1/\alpha$.

**Step 2: Spectrum of the Kronecker product matrix.** Let $(\lambda_i, u_i)$, $i = 1, \ldots, n$, be the eigenpairs of $A$ and let $(\sigma_m, v_m)$, $m = 1, \ldots, d$, be the eigenpairs of $W$. Then the eigenvalues of $\tilde{A} = W^\top \otimes A$ are exactly the products

$$\lambda_i \sigma_m, \qquad i = 1, \ldots, n, \ m = 1, \ldots, d,$$

and an associated eigenvector is $v_m \otimes u_i$. Hence,

$$\rho(A) = \max_{i,m} |\lambda_i \sigma_m| = \lambda_k\, \rho(W).$$

Thus, the linearization loses stability precisely when

$$\alpha \lambda_k\, \rho(W) = 1,$$

which proves the first part of the theorem.

**Step 3: Isolating the critical 2D structure.** At the critical point, there is at least one pair of leading eigenmodes $(k, j)$ such that

$$|\alpha \lambda_k \sigma_j| = 1 \quad \text{and} \quad \lambda_k = \lambda_k(A), \ |\sigma_j| = \rho(W).$$

Fix such a pair and set

$$u := u_k \in \mathbb{R}^n, \qquad v := v_j \in \mathbb{R}^d,$$

both normalized. We will show that the corresponding nonlinear equilibrium has the form

$$X^\star = a\,uv^\top + \text{higher-order terms},$$

with $a$ small.

**Step 4: Plugging a rank-one ansatz.** Write

$$X = a\,uv^\top + Y, \qquad \langle Y, uv^\top \rangle_F = 0,$$

i.e. $Y$ is orthogonal (in Frobenius inner product) to the main rank-one direction. Apply $A(\cdot)W$:

$$AXW = A(auv^\top + Y)W = a(Au)(v^\top W) + AYW = a(\lambda_k u)(\sigma_j v)^\top + AYW,$$

because $Au = \lambda_k u$ and $Wv = \sigma_j v$. So the "leading part" of $AXW$ is the rank-one matrix

$$a\,\lambda_k \sigma_j\,uv^\top,$$

plus the transverse contribution $AYW$.

**Step 5: Entrywise expansion of the nonlinearity.** Apply $\phi(\cdot)$ entrywise:

$$\phi(AXW) = \phi\big(a\,\lambda_k\sigma_j\,uv^\top + AYW\big) = \phi\big(a\,\lambda_k\sigma_j\,uv^\top\big) + \text{higher-order transverse terms.}$$

Since $\phi(z) = \alpha z - \frac{\gamma}{6}z^3 + O(z^5)$ entry-wise, we have for each entry $(i,j)$:

$$\phi\big(a\,\lambda_k\sigma_j\,u_iv_j\big) = a\alpha\,\lambda_k\sigma_j\,u_iv_j - \frac{\gamma(a\,\lambda_k\sigma_j)^3}{6}\,u_i^3v_j^3 + O(a^5).$$

Hence the $(i,j)$ entry of the updated matrix is

$$X'_{ij} = a\alpha\,\lambda_k\sigma_j\,u_iv_j - \frac{\gamma(a\,\lambda_k\sigma_j)^3}{6}\,u_i^3v_j^3 + \text{(terms involving } Y) + O(a^5).$$

**Step 6: Projecting back onto the main rank-one direction.** To get the scalar equation for $a$, we project onto $uv^\top$ using the Frobenius inner product:

$$\langle A, uv^\top\rangle_F = \sum_{i,j} A_{ij}u_iv_j.$$

Apply this to $X' = \phi(AXW)$, ignoring for now the transverse $Y$-terms (which are $O(a^3)$ and can be absorbed as before):

$$\langle X', uv^\top\rangle_F = \sum_{i,j}\Big(a\alpha\,\lambda_k\sigma_j\,u_iv_j - \frac{\gamma(a\,\lambda_k\sigma_j)^3}{6}\,u_i^3v_j^3\Big)u_iv_j + O(a^5)$$

$$= a\alpha\,\lambda_k\sigma_j\sum_{i,j}u_i^2v_j^2 - \frac{\gamma(a\,\lambda_k\sigma_j)^3}{6}\sum_{i,j}u_i^4v_j^4 + O(a^5).$$

Because $u$ and $v$ are unit-norm,

$$\sum_i u_i^2 = 1, \qquad \sum_j v_j^2 = 1 \quad\Longrightarrow\quad \sum_{i,j}u_i^2v_j^2 = 1.$$

Define

$$\kappa_k := \sum_i u_i^4, \qquad \xi_j := \sum_j v_j^4.$$

Then

$$\sum_{i,j}u_i^4v_j^4 = \Big(\sum_i u_i^4\Big)\Big(\sum_j v_j^4\Big) = \kappa_k\xi_j.$$

Therefore

$$\langle X', uv^\top\rangle_F = a\alpha\,\lambda_k\sigma_j - \frac{\gamma(a\,\lambda_k\sigma_j)^3}{6}\,\kappa_k\xi_j + O(a^5).$$

**Step 7: Fixed-point condition for the amplitude.** A fixed point must satisfy $X' = X$, i.e.

$$\langle X', uv^\top\rangle_F = \langle X, uv^\top\rangle_F = a.$$

Hence

$$a = a\,\alpha\lambda_k\sigma_j - \frac{\gamma(a\,\lambda_k\sigma_j)^3}{6}\,\kappa_k\xi_j + O(a^5).$$

Rearrange:

$$0 = (\alpha\lambda_k\sigma_j - 1)a - \frac{\gamma(\lambda_k\sigma_j)^3}{6}\,\kappa_k\xi_j\,a^3 + O(a^5).$$

This is the same cubic balance as in the 1D case.

**Step 8: Solving for the amplitude.** Set

$$\mu := \alpha\lambda_k|\sigma_j| - 1,$$

and assume $\mu > 0$ is small (i.e. we are just beyond the instability). Ignoring the $O(a^5)$ term,

$$(\alpha\lambda_k|\sigma_j| - 1)a = \frac{\gamma(\lambda_k\sigma_j)^3}{6}\kappa_k\xi_j\, a^3,$$

which gives

$$a^2 = \frac{6(\alpha\lambda_k|\sigma_j| - 1)}{\gamma(\lambda_k\sigma_j)^3\kappa_k\xi_j}.$$

Taking the square root yields

$$a = \pm\sqrt{\frac{6(\alpha\lambda_k|\sigma_j| - 1)}{\gamma(\lambda_k\sigma_j)^3\kappa_k\xi_j}} \; + \; O\big((\alpha\lambda_k|\sigma_j| - 1)^{3/2}\big),$$

which is exactly the amplitude stated in the theorem.

**Step 9: Rank-one structure and symmetry.** Finally,

$$X^\star_{\pm,k,\ell} = \pm a^\star_{k,\ell}\, u_k v_j^\top + O\big((a^\star_{k,\ell})^3\big)$$

comes from undoing the vectorization and restoring the transverse correction $Y = O(a^3)$, just as in the 1D proof. The two signs correspond to the odd nonlinearity and give symmetric stable branches of the supercritical pitchfork.

This completes the proof. □

## A.10. Reminders of Random Matrix Theory

We summarize the classical spectral results needed to determine the critical variance conditions in Corollary 5.2.

**Non-symmetric (Ginibre) matrices.** Let $W \in \mathbb{R}^{d\times d}$ be a non-symmetric matrix with i.i.d. real entries with mean $0$ and variance $v$, its eigenvalues are uniformly distributed in the complex disk of radius

$$R = \sqrt{dv},$$

known as the *circular law*. Therefore $\rho(W) \approx \sqrt{dv}$.

## A.11. Proof of Corollary 5.2 (bifurcation-aware initialization)

Corollary 5.2 is a direct consequence of Theorem 5.1 and the random matrix estimates recalled above.

**Step 1 (bifurcation condition).** Theorem 5.1 shows that the trivial state $X = 0$ of the layer

$$X^{(\ell+1)} = \phi(AX^{(\ell)}W)$$

loses stability when

$$\alpha\rho(W)\lambda_k = 1. \tag{31}$$

Equivalently,

$$\rho(W) = \frac{1}{\alpha\lambda_k}.$$

**Step 2 (plug in random-matrix scaling).** From the random matrix reminders A.10: $W$ is *non-symmetric* (Ginibre-type) with i.i.d. entries of variance $v$, then $\rho(W) \approx \sqrt{dv}$. Plugging this into (31) gives

$$\sqrt{dv}\,\alpha\lambda_k = 1 \quad \Longrightarrow \quad v = \frac{1}{d\alpha^2\lambda_k^2}.$$

These are exactly the two variance formulas stated in the corollary.

*Table 3.* Statistics of the benchmark datasets.

| Dataset | #Nodes | #Edges | #Features | #Classes | $h\%$ | Avg. N.D. | Diameter | Class Type |
|---|---|---|---|---|---|---|---|---|
| Cora | 2,708 | 5,278 | 1,433 | 7 | 80.4 | 3.9 | ~19 | Citation networks |
| Citeseer | 3,327 | 4,676 | 3,703 | 6 | 73.5 | 2.8 | ~16 | Citation networks |
| Pubmed | 19,717 | 44,327 | 500 | 3 | 80.2 | 4.5 | ~21 | Citation networks |
| Computers | 13,752 | 245,861 | 767 | 10 | 77.7 | 35.8 | ~14 | Co-purchase networks |
| Photo | 7,650 | 119,081 | 745 | 8 | 82.7 | 31.1 | ~12 | Co-purchase networks |
| CoauthorCS | 18,333 | 81,894 | 6,805 | 15 | 80.0 | 8.9 | ~15 | Co-authorship networks |
| Texas | 183 | 295 | 1,703 | 5 | 0.11 | 3.2 | ~9 | WebKB pages |
| Wisconsin | 251 | 466 | 1,703 | 5 | 0.21 | 3.7 | ~9 | WebKB pages |
| Cornell | 183 | 280 | 1,703 | 5 | 0.30 | 3.1 | ~8 | WebKB pages |
| Squirrel | 5,201 | 198,493 | 2,089 | 5 | 0.22 | 76.3 | ~11 | Wikipedia pages |
| Chameleon | 2,277 | 31,421 | 2,325 | 5 | 0.23 | 27.6 | ~10 | Wikipedia pages |

**Step 3 (interpretation for initialization).** Choosing the variance $v$ of the feature weight matrix $W$ at (or just above) these critical values means that already at initialization the linear part of the layer sits *on* the instability boundary:

$$\alpha\rho(W)\,\lambda_k \gtrsim 1.$$

By Theorem 5.1, this is precisely the regime where nontrivial, rank-one graph–feature patterns

$$X^\star_{\pm,k,\ell} \approx \pm a^\star_{k,\ell}\, u_k v_j^\top$$

exist and are stable. In other words, the (deep) GNN does *not* start from a fully oversmoothed, constant signal: it already has a small but structured component along the leading Laplacian eigenmode $u_k$, modulated by the leading feature direction $v_j$. This is why we refer to it as a *bifurcation-aware* initialization.

This completes the proof. $\qquad\square$

# B. Experimental details

Table 3 summarizes the key properties of the benchmark datasets used in our experiment, where $h\%$ denotes the graph homophily ratio and Avg. N.D. indicates the average node degree.

**Baselines.** We compare our approach against a diverse set of state-of-the-art baselines, including GraphSAGE (Hamilton et al., 2017), GCN (Kipf & Welling, 2017), GAT (Veličković et al., 2018), CGNN (Xhonneux et al., 2020), GRAND (Chamberlain et al., 2021), GREAD (Choi et al., 2023), and SGOS-Expn (Zhao et al., 2025). These methods cover a broad range of modelling paradigms, encompassing standard message-passing architectures (GCN, GraphSAGE), attention-based models (GAT), normalization techniques for mitigating oversmoothing (PairNorm), continuous-depth formulations based on neural ODEs and diffusion dynamics (CGNN, GRAND, GREAD), as well as recent spectral optimization approaches (SGOS-Expn). All baseline results are taken from Zhao et al. (2025) to ensure a fair comparison in identical experimental settings and data splits.

**Hyperparameter Settings.** We optimize hyperparameters for each model using random search over 50 configurations. The search space includes a hidden dimension in $\{64, 128, 256\}$, dropout rate in $[0, 0.7]$, learning rate in $[10^{-3.5}, 10^{-1.5}]$, weight decay in $[10^{-5}, 10^{-1.5}]$, network depth $\in \{2, 3, 4, 5\}$, and polynomial filter order $K \in \{1, 2, 3\}$. We select the configuration with the highest validation accuracy and report the corresponding test performance. Table 4 provides the optimal network depths and polynomial filter order $K$ for each dataset and for the proposed architectures.

**Training.** Models are trained using the Adam optimizer with early stopping and a patience of 100 epochs, where validation performance is evaluated every 10 epochs. For all datasets, we adopt random $60\%/20\%/20\%$ train/validation/test splits and report the mean $\pm$ standard deviation over 10 runs.

**Implementation.** All experiments are conducted on a server equipped with NVIDIA H100 GPUs (CUDA 12.6) with 95 GB of memory, enabling efficient training and evaluation on high-complexity graphs. The implementation and related resources will be made publicly available upon acceptance of the paper.

*Table 4.* **Best hyperparameters (depth, $K$) for polynomial-$A$ models.**

| Method | Texas | Wisconsin | Squirrel | Chameleon | Cornell | Computer | CiteSeer | PubMed | Cora | CoauthorCS | Photo |
|---|---|---|---|---|---|---|---|---|---|---|---|
| Sine-Poly(A)-sh | $(3,2)$ | $(3,2)$ | $(4,1)$ | $(4,1)$ | $(3,2)$ | $(4,2)$ | $(2,2)$ | $(4,2)$ | $(3,1)$ | $(2,2)$ | $(3,2)$ |
| Sine-Poly(A)-pl | $(4,2)$ | $(3,2)$ | $(4,1)$ | $(4,2)$ | $(4,2)$ | $(4,1)$ | $(2,2)$ | $(3,2)$ | $(3,2)$ | $(3,2)$ | $(3,2)$ |
| ReLU-Poly(A)-sh | $(4,2)$ | $(2,1)$ | $(2,1)$ | $(2,1)$ | $(2,2)$ | $(3,1)$ | $(2,2)$ | $(3,2)$ | $(3,2)$ | $(3,1)$ | $(3,2)$ |
| ReLU-Poly(A)-pl | $(4,1)$ | $(3,2)$ | $(3,1)$ | $(3,1)$ | $(2,2)$ | $(2,2)$ | $(3,2)$ | $(3,2)$ | $(4,2)$ | $(4,2)$ | $(3,1)$ |

*Table 5.* **Average rank ($\pm$ std) across all benchmark datasets.** Methods are ranked by accuracy on each dataset (1 = best), and the average rank is computed across all datasets except large-scale OGBN-arXiv. Lower is better.

| Method | Mean Rank |
|---|---|
| Sine-Poly(A)-sh | $3.00 \pm 2.13$ |
| Sine-Poly(A)-pl | $3.18 \pm 3.31$ |
| GELU-Poly(A)-pl | $4.82 \pm 1.78$ |
| GELU-Poly(A)-sh | $5.32 \pm 2.56$ |
| ReLU-Poly(A)-pl | $6.64 \pm 2.41$ |
| ReLU-Poly(A)-sh | $6.91 \pm 3.56$ |
| Swish-Poly(A)-pl | $7.36 \pm 2.26$ |
| Swish-Poly(A)-sh | $7.59 \pm 2.35$ |
| SGOS-Expn | $7.73 \pm 4.76$ |
| GREAD | $8.45 \pm 4.46$ |
| Sine GCN | $10.73 \pm 6.08$ |
| GraphSAGE | $12.00 \pm 2.65$ |
| GRAND | $12.45 \pm 3.70$ |
| GCN | $12.91 \pm 1.97$ |
| CGNN | $13.45 \pm 1.81$ |
| GAT | $14.82 \pm 1.60$ |
| PairNorm | $15.13 \pm 2.17$ |

*Table 6.* **Activation ablation.** Node classification accuracy (%) on benchmark datasets without polynomial filtering and for different base activation functions. GELU and Swish alternatives are added. Results are reported as mean $\pm$ standard deviation across multiple runs. Datasets are ordered by increasing homophily levels (Hom.), where higher values indicate more homophilic graphs.

| Method | Texas | Wisconsin | Squirrel | Chameleon | Cornell | Citeseer | Computers | Pubmed | Cora | CoauthorCS | Photo |
|---|---|---|---|---|---|---|---|---|---|---|---|
| *Hom. level* | *0.11* | *0.21* | *0.22* | *0.23* | *0.30* | *0.74* | *0.78* | *0.80* | *0.81* | *0.81* | *0.83* |
| Sine-Poly(A)-sh | $90.09 \pm 0.1$ | $85.62 \pm 0.0$ | $60.81 \pm 0.56$ | $70.81 \pm 0.24$ | $76.58 \pm 0.1$ | $78.18 \pm 1.01$ | $91.56 \pm 0.32$ | $89.71 \pm 0.09$ | $89.78 \pm 0.20$ | $95.84 \pm 0.10$ | $95.69 \pm 0.17$ |
| Sine-Poly(A)-pl | $93.69 \pm 0.2$ | $80.39 \pm 0.0$ | $61.79 \pm 0.40$ | $70.85 \pm 0.92$ | $76.87 \pm 0.1$ | $78.15 \pm 0.75$ | $91.94 \pm 0.07$ | $88.67 \pm 0.27$ | $89.89 \pm 0.50$ | $95.42 \pm 0.07$ | $95.55 \pm 0.21$ |
| Sine GCN | $52.68 \pm 1.19$ | $50.57 \pm 2.77$ | $29.25 \pm 0.55$ | $44.27 \pm 1.47$ | $40.50 \pm 2.92$ | $78.12 \pm 0.38$ | $91.25 \pm 0.10$ | $88.38 \pm 0.18$ | $89.85 \pm 0.19$ | $94.15 \pm 0.12$ | $94.50 \pm 0.35$ |
| ReLU-Poly(A)-sh | $92.79 \pm 0.1$ | $86.27 \pm 0.1$ | $53.57 \pm 0.82$ | $69.63 \pm 0.35$ | $67.57 \pm 0.2$ | $77.56 \pm 0.55$ | $91.80 \pm 0.43$ | $88.36 \pm 0.14$ | $89.36 \pm 0.18$ | $93.42 \pm 0.14$ | $94.37 \pm 0.25$ |
| ReLU-Poly(A)-pl | $90.09 \pm 0.1$ | $84.31 \pm 0.0$ | $53.67 \pm 0.46$ | $69.89 \pm 0.49$ | $76.58 \pm 0.1$ | $77.53 \pm 0.55$ | $91.85 \pm 0.26$ | $88.46 \pm 0.19$ | $89.47 \pm 0.21$ | $93.33 \pm 0.19$ | $94.09 \pm 0.27$ |
| GELU-Poly(A)-sh | $90.09 \pm 1.27$ | $83.65 \pm 0.89$ | $53.74 \pm 0.44$ | $67.61 \pm 0.27$ | $77.47 \pm 3.01$ | $77.74 \pm 0.25$ | $92.62 \pm 0.19$ | $88.77 \pm 0.24$ | $88.87 \pm 0.68$ | $93.51 \pm 0.06$ | $94.88 \pm 0.31$ |
| GELU-Poly(A)-pl | $91.89 \pm 1.15$ | $85.53 \pm 1.78$ | $54.77 \pm 0.83$ | $67.32 \pm 0.18$ | $76.57 \pm 3.59$ | $77.60 \pm 0.07$ | $92.38 \pm 0.43$ | $88.85 \pm 0.16$ | $89.11 \pm 0.52$ | $93.50 \pm 0.07$ | $95.06 \pm 0.24$ |
| Swish-Poly(A)-sh | $89.18 \pm 2.34$ | $84.97 \pm 1.02$ | $54.25 \pm 0.46$ | $67.18 \pm 0.54$ | $67.56 \pm 2.24$ | $77.25 \pm 0.25$ | $91.76 \pm 0.69$ | $88.77 \pm 0.07$ | $88.93 \pm 0.67$ | $93.41 \pm 0.07$ | $94.99 \pm 0.14$ |
| Swish-Poly(A)-pl | $90.09 \pm 1.07$ | $83.02 \pm 4.08$ | $53.61 \pm 0.16$ | $67.97 \pm 0.53$ | $73.67 \pm 3.25$ | $77.55 \pm 0.14$ | $92.51 \pm 0.62$ | $88.66 \pm 0.07$ | $88.87 \pm 0.52$ | $93.37 \pm 0.02$ | $94.80 \pm 0.05$ |

