# OpenReview forum: "Beyond ReLU: Bifurcation, Oversmoothing, and Topological Priors"
_ICML.cc/2026/Conference — ICML 2026 spotlight_

### Official Review · Reviewer_qTqi · 2026-02-23

**Soundness:** 2
**Presentation:** 2
**Significance:** 2
**Originality:** 3
**Overall Recommendation:** 4
**Confidence:** 3

**Summary:**

This paper first re-frame the oversmooth in graph neural networks from bifurcation theory perspective. By theoretical proving that the representation collapse is actually a monostability point of the system, the authors proposed new activation functions to avoid this. Their theory was proven in both toy convergence experiments, depth-robustness experiments on Cora, and node classification benchmarks.

**Compliance With Llm Reviewing Policy:**

Affirmed.

**Final Justification:**

Since my concerns has been mostly solved, I raised my recommendation to weak accept.

**Key Questions For Authors:**

- [Q1] Under what realistic architectural assumptions does the claimed contractivity of standard ReLU-based GNN updates hold?
- [Q2] Does the bifurcation analysis formally apply only locally near the origin, or does it imply global guarantees?
- [Q3] What mathematical justification supports the reduction of nonlinear dynamics to a single eigenvector direction in Theorem 3.2?
- [Q4] How is the scalar bifurcation condition rigorously extended to the matrix setting in Theorem 5.1?
- [Q5] What is the formal connection between the infinite-depth fixed-point analysis and the behavior of finite-depth trained networks?
- [Q6] Could the authors provide full implementation details for the 64-layer experiments?
- [Q7] Have other smooth activations (e.g., GELU/Swish) been tested, and do they exhibit similar behavior?

**Limitations:**

YES

**Strengths And Weaknesses:**

## Strength

- [S1] The theoretical viewpoint linking oversmoothing to bifurcation instability is intellectually appealing and potentially impactful. The work focused on the interpretation of representational collapse as convergence toward a homogeneous stable fixed point in a nonlinear dynamical system. The derivation of a pitchfork bifurcation condition and the associated amplitude scaling near criticality (Theorem 3.2) provides an explicit analytical connection between activation derivatives, spectral graph properties, and representation stability.
- [S2] The attempt to translate the nonlinear stability analysis into a graph-aware initialization scheme and demonstrating empirical depth robustness. This effort to connect theoretical nonlinear dynamics to concrete initialization design is creditable.
- [S3] The core hypothesis that activation nonlinearities influence the stability landscape of GNN propagation, rather than merely their expressivity, is interesting and could motivate further theoretical work on stability-aware architectural design.

## Weakness

- [W1] Several central claims rely on assumptions that are either unstated or unrealistic for practical GNNs. (1)  **Contractivity of standard GNNs**. The argument that ReLU-based GNN updates are contractive requires $||A|| \cdot ||W|| \cdot \sup|\phi'| < 1$, where the normalized adjacency $||A|| = 1$ and the ReLU satisfy $\sup|\phi'|=1$. So it actually requires $||W||<1$. But the trained networks often have spectral norms larger than 1, and residual connections will explicitly break the contraction. (2) **The Taylor expansion**. In the paper, the expansion $\phi (x) \approx \alpha x - \beta x^2 - \gamma x^3 \cdots$ uses local polynomial approximation near $x=0$. This is only correct when $|x| << 1$. But fixed points may occur at large amplitude, and nonlinear dynamics may leave the local neighborhood. Thus, this is only a local bifurcation near zero and is not proper to interpret as global GNN behavior. (3) **Eigenvector dominance assumption**. In the theorem 3.2, the authors stated that for $w > w_k$, $x_{\pm}^{\*}  = \pm a^{\*} (w) u_k + O((a^{\*})^3)$. This implicitly assumes that nonlinear dynamics can be reduced to 1D subspace, and couple to other eigenmodes is negligible. But nonlinear operator $\phi (wAx)$ does not preserve eigenspaces. Thus there is a missed center-manifold reduction or invariant-subspace argument. (4) Theorem 5.1 extends scalar model to $X^{l+1}=\phi (AX^{(l)}W)$. But here the scalar weight $w$ is replaced directly by spectral radius $\rho (W)$, which is not generally valid. A proof includes joint spectral decomposition, tensor-product eigenbasis analysis and perturbation analysis of nonlinear operator would be needed. (5) The theoretical framework analyzes asymptotic convergence of the nonlinear iteration, whereas real GNNs are trained with finite depth and gradient-based optimization. The link between fixed-point bifurcation behavior and practical training dynamics is not formally established.
- [W2] **Experimental section**. The current section does not fully support the theoretical claims. For instance, the synthetic convergence experiments validated the properties of a fixed nonlinear iteration rather than trained neural networks. This confirms mathematical predictions but does not demonstrate improved learning behavior. In addition, the reported ability to train 64-layer GNNs depends on details such as architecture details, normalization methods, and regularization choices, which are not reported. As for the node classification benchmarks, they mostly involve shallow networks where oversmoothing is not the dominant issue. The authors acknowledge this, which weakens the empirical support for the theoretical motivation. A complete ablation experiment would also be required, including comparison with other smooth activations (e.g., GELU, Swish), sensitivity to cubic coefficient magnitude and comparison with standard deep-GNN stabilization methods.
- [W3] **Presentation and clarity**. In the Corollary 3.5, the last sentence of supercritical regime is **unfinished**. The organization of the theoretical part can be improved, for example, the narrative jumps between fixed-point theory, energy landscapes, NTK analysis and spectral filtering without a clearly staged structure. Extensive physics analogies provide intuition but occasionally obscure mathematical assumptions. And the key geometric mechanism behind eigenvector alignment deserves clearer intuitive explanation before formal theorem statements.

---

> ### Author Rebuttal · Authors · 2026-03-31
>
> We sincerely thank the reviewer for their rigorous and insightful evaluation. We are encouraged that you find our theoretical viewpoint linking oversmoothing to bifurcation instability intellectually appealing, and our core hypothesis, that activation non-linearities govern the stability landscape, interesting. Your detailed technical critique has helped us identify exactly where our presentation needs strengthening. We address your concerns below.
>
> **[Q1] Under what realistic architectural assumptions does the claimed contractivity of standard ReLU-based GNN updates hold?**
>
> Our argument after Lemma 3.1 was to highlight that, as identified in (Arroyo et al., 2025), oversmoothing can be seen as a contractivity issue. However, as shown in the rest of our paper (e.g., Theorem 3.5), simply having a non-contractive coupling |W|>1 is insufficient to avoid oversmoothing. We will clarify this in the final version.
>
> **[Q2] Does the bifurcation analysis formally apply only locally near the origin, or does it imply global guarantees?**
>
> We chose $ x=0 $ pedagogically because it is the trivial fixed point associated with representational collapse, making subsequent derivations simpler. Nevertheless, in dynamical systems theory, the stability analysis around an arbitrary large fixed point can be transformed (with a change of coordinates) into a study around $0$. In the revised version, we will add a remark detailing that our bifurcation analysis is indeed applicable to arbitrary fixed point values $x^*$.
> While dynamics *can* leave the local neighborhood, bounding the behavior near the unstable equilibrium is a standard dynamical systems approach (tracing back to Poincaré's work on the stability of dynamical systems) that provides rigorous sufficient conditions for instability.
>
> **[Q3] What mathematical justification supports the reduction of nonlinear dynamics to a single eigenvector direction in Theorem 3.2?**
>
> We thank the reviewer for this technical question. To clarify this property: we do not assume the non-linear operator preserves eigenspaces. In our proof (Appendix A.3, Step 3), the Taylor expansion around fixed point $x^*$ justifies this local preservation of the eigenspace. Specifically, we rigorously show that the mode coupling terms are negligible up to the 3rd order. While this is indeed a local argument, its predictive power is strongly validated in practice. It is verified by the empirical scaling laws matching our exact theoretical predictions in the synthetic setting, but also in the learning setting represented by Figure 5. We will explicitly discuss the local nature of the mode-coupling reduction in the revised version.
>
> **[Q4] How is the scalar bifurcation condition rigorously extended to the matrix setting in Theorem 5.1?**
>
> You correctly pointed out that replacing the scalar weight with the spectral radius requires a rigorous joint spectral decomposition, tensor-product eigenbasis analysis, and perturbation analysis of the non-linear operator. We actually provide exactly this proof in Appendix A.8. Thanks to your comment, we noted that this specific appendix section was not referenced near Theorem 5.1, and have corrected this in the camera-ready version.
>
> **[Q5] What is the formal connection between the infinite-depth fixed-point analysis and the behavior of finite-depth trained networks?**
>
> This is a fair critique. In the contractive regime, controllable error bounds can be established via similar arguments used to prove Banach's theorem.
>
> **[Q6] Could the authors provide full implementation details for the 64-layer experiments?**
>
> Please note that in our experiments with a 64-layer GNN we used a completely vanilla GCN:  *no* residual connections and *no* layer/batch normalization were applied. The only modifications were the swap of the activation function and our bifurcation-aware initialization. The fact that the network trains successfully to 64 layers relying *solely* on the activation dynamics is the strongest empirical validation of our theory. We apologize for omitting these vital architectural constraints in the text and have explicitly stated them in the experimental setup in Appendix B.
>
> **[W2]: Ablation**
> To improve our ablation experiments, we include PairNorm , GELU-Poly(A) and SWISH-Poly(A). Here is the mean rank of these methods (will be added in the revised version):
> | Method | Mean Rank ± Std |
> |--------|-----------------|
> | Sine-Poly(A)-sh | 3.00 ± 2.13 |
> | GELU-Poly(A)-sh | 5.32 ± 2.56 |
> | Swish-Poly(A)-sh | 7.59 ± 2.35 |
> | PairNorm | 15.12 ± 2.17 |
>
> **[W3]: Presentation and clarity**
> We will ensure that all comments are incorporated into the final version and strongly believe that they can be made within a minor revision. Additionally, we will release a complete implementation of our method to ensure full reproducibility of all of our results and to enable follow-up work.

---

> > ### Author Rebuttal · Reviewer_qTqi · 2026-04-03
> >
> > I thank the authors for the response. I will adjust my recommendation accordingly.

---

> > > ### Author Response · Authors · 2026-04-05
> > >
> > > We thank the reviewer for their positive response and for highlighting in their acknowledgement that their concerns were fully resolved.
> > >
> > > Since the reviewer mentioned that they will adjust their score, and it seems the score in OpenReview is still the same as pre-rebuttal, we would respectfully ask the reviewer to consider updating their Overall Recommendation.
> > >
> > > We would be happy to answer any additional questions or comments. Thank you for your valuable feedback and for helping us to make this work better!

---

### Official Review · Reviewer_fdi1 · 2026-03-09

**Soundness:** 2
**Presentation:** 3
**Significance:** 3
**Originality:** 3
**Overall Recommendation:** 4
**Confidence:** 3

**Summary:**

This paper reframes oversmoothing in GNNs as a dynamical stability problem, formalized through bifurcation theory. The central claim is that standard ReLU activation places the network in a regime where the only stable fixed point of the layer-wise iteration is the trivial homogeneous state, whereas replacing it with activations satisfying σ'(0) > 0 and σ'''(0) < 0 (e.g., sin, tanh) induces a pitchfork bifurcation that destabilizes homogeneity and creates a pair of stable, non-trivial fixed points. The authors introduce an effective potential V(x) whose landscape encodes this stability structure, derive precise amplitude scaling laws for the bifurcated states, and show that near-critical initialization also creates a rank-one NTK that biases learning toward a specific graph eigenmode. A closed-form bifurcation-aware initialization scheme is derived and validated on depth robustness experiments on a node classification benchmark across eleven datasets.

**Compliance With Llm Reviewing Policy:**

Affirmed.

**Key Questions For Authors:**

Lines 154–158 state that fixed points are stationary points of V, with stable fixed points at local minima and unstable ones at local maxima. What is the theoretical basis for this correspondence? Specifically, why do local minima of V correspond to stable fixed points and local maxima to unstable ones?

**Limitations:**

yes

**Strengths And Weaknesses:**

## Strengths

**S1. Rigorous and insightful theoretical framing.** The connection between activation function properties and bifurcation-induced pattern formation is well-grounded. The main theorem gives concrete, checkable conditions on the activation function and derives testable scaling laws that are verified experimentally across multiple graph topologies. This is a meaningful theoretical advance over purely empirical activation function comparisons.

**S2. Clean theoretical progression.** The paper builds from a simplified scalar GNN setting, through a NTK interpretation that connects near-critical initialization to topological learning priors, and then to realistic multi-feature GNN architectures. The use of an effective potential to translate fixed-point stability into an energy landscape is particularly effective for building geometric intuition.

**S3. Practical output: closed-form initialization.** The bifurcation analysis directly yields an explicit initialization formula that places the network at criticality. This is actionable and directly tested in Figure 5, which shows convincing depth robustness up to 64 layers for bifurcating activations versus rapid collapse for ReLU.

## Weaknesses

**W1. The experimental design does not isolate the contribution of activation function replacement.** The proposed models combine two changes simultaneously: replacing the activation function and introducing a learnable polynomial filter. The baseline methods in Table 1 use standard message passing with their original activations. It is therefore unclear whether performance gains stem from the theoretically motivated activation substitution, the polynomial filter, or their combination. The most direct test of the paper's central claim would be replacing the activation function in an existing strong baseline while keeping all other components fixed. Without this comparison, the experiments do not cleanly validate the theory. Additionally, a monomial filter variant would help establish a performance lower bound and clarify how much of the gain is attributable to the polynomial filter design rather than the activation function.

**W2. The paper is difficult to read for non-specialists.** The theoretical core of the paper draws heavily on dynamical systems concepts (bifurcation theory, Lyapunov-Schmidt reduction, effective potentials, and the neural tangent kernel) without providing sufficient background for a general machine learning audience. Key concepts are introduced with minimal explanation, and the connection between the mathematical formalism and the practical problem of oversmoothing is not always made explicit at each step. The paper would benefit from a clearer expository thread that guides the reader through the chain of reasoning, particularly in Sections 3 and 4.

---

> ### Author Rebuttal · Authors · 2026-03-31
>
> We sincerely thank Reviewer fdi1 for their thorough and encouraging review. We are grateful that the reviewer found the theoretical framing rigorous, the progression clean, and the closed-form initialization practically valuable.
> In the following, we clarify the experimental design choices you highlighted and provide the theoretical justification for the potential energy landscape.
>
> ###  Weaknesses
> **W1: Table 1 combines the new activation with a polynomial filter, making it unclear if performance gains stem from the activation, the filter, or both. An ablation is needed.**
>
> Thank you for highlighting this. We acknowledge that our exposition here may have blurred the lines between two complementary analyses. We will clarify this distinction in the revised text.
>
> - Figure 5 is an isolation test for oversmoothing. In these experiments, we use a standard baseline GCN architecture and only swap the activation function and initialization. We observe in this setting that replacing ReLU with our proposed activations already prevents oversmoothing up to 64 layers, which cleanly proves our central claim.
> - Table 1 serves a different purpose: to test topological bias selection in shallow regimes. As shown in Theorem 5.1 and Corollary 4.1, the bifurcation mechanism naturally biases the network toward the dominant eigenmode of the adjacency matrix (typically low-frequency features). While this inductive bias is useful for some tasks (e.g., on the Cora dataset), other tasks may require different frequencies, justifying the use of polynomial filtering.
>
> We emphasize that Table 1 was designed to show that our activation functions do not cripple the network’s expressivity in standard shallow tasks, and that they pair successfully with filters when bias selection is necessary. We also fully agree that adding a pure ``Baseline GCN + Sine Activation'' to Table 1 will help establish a clearer lower bound. Experiments indicate that its mean rank is at **7.82**, with weak results on heterophilic datasets. We will add an ablation in Table 1 showing these full results. Additionally, we will perform a similar experiment to Figure 5 on a heterophilic dataset.
>
> **W2: The paper draws heavily on dynamical systems concepts without sufficient background for a general ML audience.**
>
> We deeply value this feedback. Our goal is to provide valuable tools to the ML community, and it requires a digestible presentation to guide the reader through the main derivations! Therefore, we will use the extra page allowed in the final version to add a gentler, more intuitive introduction to Lyapunov-Schmidt reduction and bifurcation theory, ensuring the connection between the mathematical formalism and practical GNN design is clearly exposed at every step. We will also state our *main results* without using dynamical systems terminology to make the key conclusions understandable by the wider community.
>
> ###  Question
> **Lines 154-158 state that fixed points are stationary points of $V$, with stable fixed points at local minima and unstable ones at local maxima. What is the theoretical basis for this correspondence?**
>
> We view the iterative layer update as a discrete gradient descent step on the effective potential $V(x)$. From Equation 3, we defined the potential such that
> $$-V'(x) = \phi(wx) - x$$
>
> This means the layer update fundamentally pushes the state $x$ in the direction of the negative gradient of $V$:
> $$x^{(l+1)} - x^{(l)} = -V'(x^{(l)})$$
>
> To rigorously illustrate why local minima are stable and maxima are unstable, it is helpful to look at the continuous-time analog of this system:
> $$\frac{dx}{dt} = -V'(x)$$
>
> Let $x^{\*}$ be a fixed point, meaning $V'(x^{\*}) = 0$. If we introduce a small perturbation $\delta x(t) = x(t) - x^{\*}$ and take the second-order Taylor expansion of the potential around $x^*$, we get:
> $$V(x) \approx V(x^{\*}) + V'(x^{\*})\delta x + \frac{1}{2}V''(x^{\*})(\delta x)^2$$
>
> Since $V'(x^{\*}) = 0$, the gradient of the potential near the fixed point is linearly approximated by $V'(x) \approx V''(x^{\*})\delta x$. Substituting this into our continuous-time dynamics yields the linear ordinary differential equation:
> $$\frac{dx}{dt} = \frac{d(\delta x)}{dt} = -V''(x^{\*})\delta x$$
>
> The solution to this ODE governs the behavior of the perturbation over time:
> $$\delta x(t) = \delta x(0) e^{-V''(x^*)t}$$
>
> This explicitly demonstrates the stability criterion based on the curvature $V''(x^*)$:
>
> - Local Minimum ($V''(x^{\*}) > 0$): The exponent $-V''(x^{\*})$ is negative, causing the perturbation to **decay exponentially** to zero. The system returns to $x^*$, making it a **stable** fixed point.
> - Local Maximum ($V''(x^{\*}) < 0$): The exponent $-V''(x^{\*})$ is positive, causing the perturbation to **diverge exponentially**. The system is pushed away from $x^*$, making it an **unstable** fixed point.
>
> We will add this continuous-time expansion to the final version to formally ground the energy landscape intuition.

---

> > ### Author Rebuttal · Reviewer_fdi1 · 2026-04-04
> >
> > I thank the authors for their detailed response. After reading the rebuttal and the other reviewers' comments, I believe that my original assessment is consistent with the current quality of the paper.

---

> > > ### Author Response · Authors · 2026-04-05
> > >
> > > We thank the reviewer for their feedback and for acknowledging that their questions have been fully addressed.

---

### Official Review · Reviewer_beFe · 2026-03-10

**Soundness:** 3
**Presentation:** 3
**Significance:** 3
**Originality:** 3
**Overall Recommendation:** 5
**Confidence:** 3

**Summary:**

The paper argues that the phenomenon of oversmoothing in GNNs can be avoided by choosing a suitable activation function that creates a bifurcation which induces novel non-trivial fixed points for the dynamics of the GNN. Furthermore, this work establishes and tests precise scaling laws for Dirichlet energy and amplitude with respect to activation slope and coupling strength. It is also suggested that this bifurcation functions as an inductive bias that speeds up learning through a NTK argument. The authors perform several experiments empirically validating the fact that Dirichlet energy and test accuracy degrade with depth when ReLU activations are used and the performance of different activations and standard methods on node classification tasks.

**Compliance With Llm Reviewing Policy:**

Affirmed.

**Final Justification:**

My main concerns were about notational inaccuracies and missing details for reproducibility. I have increased my score to 5 following the rebuttal which addressed all of my concerns.

**Key Questions For Authors:**

1. The graph Laplacian operator $L$ is used on lines 660 (although I think it should be $A$ here) and 985 to state that $Lu_k = \lambda_k u_k$, where $\lambda_k$ and $u_k$ are the eigenpairs for the adjacency $A$. While the normalized adjacency matrix and the laplacian could have related eigenpairs, I do not see why they have to match?

2. On line 435 you mention that the networks used for node classification are shallow, but as far as I can tell you never mention how many layers you end up using in Sine-Poly(A) and ReLU-Poly(A). I get that you sweep between {2,3,4,5}, but I feel that it is relevant to mention the final choice. The same applies to the polynomial filter order.

3. I believe there are a few notational mistakes, for example in equation 4 it looks like you absorb the factorial terms of the Taylor expansion in the coefficients $\alpha, \beta, \gamma$ but then in equation 5 those terms reappear which makes the two inconsistent. Another example is the sign of the cubic term in line 658.

**Limitations:**

Yes

**Strengths And Weaknesses:**

**Strengths**
- The paper is well written and the introduction of a simple 1D model from which to generalize helps a lot with readability.
- The relation between the degrees of the Taylor expansion and the bifurcation properties they induce are insightful and general.
- The figures are clean, pretty and help get the point they are making effectively.
- The empirical validation is convincing and thorough.

**Weaknesses**
- While the proofs I managed to check seem correct, there are several notational errors that make the reasoning harder to follow.
- Some details about the experimental validation are lacking, which makes future reproducibility more difficult.

---

> ### Author Rebuttal · Authors · 2026-03-31
>
> We thank Reviewer beFe for their assessment and for highlighting the clarity, insightfulness, and empirical strength of our work. We are especially grateful that the reviewer appreciated our exposition, building from a simple 1D model to introduce the relevant tools and predictions. Our goal with the effective energy plots was precisely to provide an intuitive geometric picture of how coefficients impact the stability environment.
>
> We also appreciate your careful reading of the mathematical proofs and experimental setup. Thank you for highlighting several notational inconsistencies and omissions, which we address below.
>
> ###  Question 1: Laplacian vs. Adjacency Operator
>
> Indeed! They do not inherently match, and we do not have to make this assumption. This is an artifact of our earlier drafts, in which we transitioned to expressing everything in terms of the normalized adjacency matrix A (standard in the GNN literature) rather than the Laplacian L.
> We will use the adjacency matrix $A$ (and its eigenpairs) and ensure the notation and exposition are mathematically consistent in the final version.
>
> ### Weakness 2, Question 2: Missing Experimental Details (Reproducibility)
>
> We agree that these details are critical for reproducibility. Here are the optimal hyperparameters for the Sine-Poly(A) and ReLU-Poly(A) models across the datasets:
>
> ### Best hyperparameters (depth, K) for polynomial-A models
>
> | Method | Texas | Wisconsin | Squirrel | Chameleon | Cornell | Computer | CiteSeer | PubMed | Cora | CoauthorCS | Photo |
> |--------|-------|-----------|----------|-----------|---------|----------|----------|--------|------|------------|-------|
> | Sine-Poly(A)-sh | (3,2) | (3,2) | (4,1) | (4,1) | (3,2) | (4,2) | (2,2) | (4,2) | (3,1) | (2,2) | (3,2) |
> | Sine-Poly(A)-pl | (4,2) | (3,2) | (4,1) | (4,2) | (4,2) | (4,1) | (2,2) | (3,2) | (3,2) | (3,2) | (3,2) |
> | ReLU-Poly(A)-sh | (4,2) | (2,1) | (2,1) | (2,1) | (2,2) | (3,1) | (2,2) | (3,2) | (3,2) | (3,1) | (3,2) |
> | ReLU-Poly(A)-pl | (4,1) | (3,2) | (3,1) | (3,1) | (2,2) | (2,2) | (3,2) | (3,2) | (4,2) | (4,2) | (3,1) |
>
> We have added a comprehensive table detailing the optimal hyperparameters for every dataset and model variant in Appendix B of the revised manuscript. Since our code will be released publicly, providing full experimental details is natural to reproduce our results and will enable follow-up work.
>
> ### Weakness 1, Question 3: Notational Mistakes
>
> Thank you for pointing this mistakes. Here are the following typos/notational errors that will be corrected in the final version:
> -  Figure 1: $\alpha w<1, \alpha w>1$ instead of $w<1, w>1$
> - Equation 4: added factorial terms to the expansion: $\phi(x) \approx \alpha x - \frac{\beta}{2} x^2 - \frac{\gamma}{6} x^3 + \dots$ instead of $\phi(x) \approx \alpha x - \beta x^2 - \gamma x^3 + \dots$.
> - Line 658: changed the sign of cubic term to negative: $-\frac{\gamma}{6} a^3$ instead of $\frac{\gamma}{6} a^3$.
> - Line 660: $A$ instead of $L$.
> - Lines 985 to 1009: $A$ instead of $L$.

---

> > ### Author Rebuttal · Reviewer_beFe · 2026-04-03
> >
> > Thank you for the rebuttal. Given the commitment of the authors to improve the discussed inaccuracies in the text I have increased my score.

---

> > > ### Author Response · Authors · 2026-04-05
> > >
> > > We thank deeply the reviewer for their positive response and for increasing their score.
> > > Your thorough review helped us improve the quality and soundness of our work!

---

### Official Review · Reviewer_fKq6 · 2026-03-11

**Soundness:** 2
**Presentation:** 3
**Significance:** 3
**Originality:** 3
**Overall Recommendation:** 4
**Confidence:** 4

**Summary:**

the paper studies the oversmoothing problem in GNN from bifurcation theory perspective.The paper proposes undesired stability can be broken by replacing standard monotone activations (e.g., ReLU) with a class of functions. Empirical validation showing deep GNNs  remain stable under these activations while ReLU-based models collapse

**Compliance With Llm Reviewing Policy:**

Affirmed.

**Final Justification:**

i've read all the author rebuttal, and will keep my rating as weak accept

**Key Questions For Authors:**

1. can we get results on some large-scale graphs (such as OGB)?
2. Does the theory extend to attention-based GNNs (GAT)?

**Strengths And Weaknesses:**

strength:
1.the paper is well written and easy to follow
2.bifurcation theory for GNN analysis is novel

weakness:
1.the theoretical bifurcation analysis assumes fairly simple model structure (7), which is far more simple than the actual GNN. the theory is still restrictive
2. the experiment performance is mixed.

---

> ### Author Rebuttal · Authors · 2026-03-31
>
> We sincerely thank Reviewer fKq6 for their thoughtful feedback and for recognizing the novelty of applying bifurcation theory to GNN analysis. We are especially glad the reviewer appreciated our efforts in presentation; our primary goal was to make these concepts from physics and dynamical systems intuitive and actionable for practitioners.
>
> Below, we address your specific concerns and questions.
>
> **The theoretical bifurcation analysis assumes a fairly simple model structure (Eq 7), which is far simpler than actual GNNs.**
>
> We agree that Equation 7 is a simplified 1D-feature GCN. However, this simplification is strictly pedagogical to clearly illustrate the bifurcation phenomena. We structured the paper in a way that unrolls the results progressively to build intuition before introducing the full complexity. Crucially, the theory is not restricted to this 1D case. As detailed in
> Section 5 and Theorem 5.1 (and fully expanded in Theorem A.1), we explicitly extend the analysis to the realistic, multi-dimensional GCN update with random weight initialization. All our theoretical results, including the bifurcation conditions and the emergence of stable rank-one patterns, hold in this realistic setting and provide actionable GCN modifications (activation replacement and bifurcation-aware initialization). Nevertheless, we acknowledge that this formulation does not include engineered architectures (eg. residual connections, graph rewiring) nor sophisticated modeling (eg. Physics inspired models such as GRAND). Although this could be an interesting extension of our work, our goal was to prove that oversmoothing is not inherent to standard GCNs and can be mitigated with proper tuning of its learning dynamics.
>
>
> **[W2] The experiment performance is mixed.**
>
>
> We understand your concern regarding the quantitative relevance of the benchmark experiments. We would like to clarify the dual purpose of our experimental section:
> - Deep Regime Validation (Figure 5): The primary goal of our experiments is to validate our theoretical claims regarding oversmoothing. By testing vanilla GCN architectures with very deep networks (64 layers), we demonstrate that our activation switch and initialization prevent collapse, whereas standard ReLU networks fail.
> - Shallow Regime Competitiveness (Table 1): Because most standard node classification tasks perform well with shallow networks, it was important to ensure that our proposed activation switch does not hinder performance in these standard regimes. To clarify the results obtained, we have added the mean rank over the datasets for each method, along with addtional ablations (Sine+GCN, GELU+Poly(A), SWISH+Poly(A)) . Specifically, Sine+Poly(A)-sh has the highest mean rank (**2.14**), closely followed by Sine+Poly(A)-pl (**2.36**), then by their ReLU-based counterparts (**4.18/4.77**). The proposed architecture outperforms recent competitor SGOS (mean rank of 5.09) on 9 out of 11 datasets. These results indicate the overall superior performance of the proposed methods.
>
>
> **[Q1] Large-scale graphs (OGB)?**
>
> We thank the reviewer for suggesting an evaluation on large-scale graphs. We have conducted experiments on the OGBN-Arxiv and OGBN-Products datasets under the standard OGB evaluation protocol.
> Across all four variants of our model, we achieve up to **74\%** accuracy on OGBN-Arxiv. This performance is comparable to strong baselines; for example, on OGBN-Arxiv, it outperforms GREAD (72\%), GRAND (72\%), GCN (72\%), and GAT (73\%) reported in the literature under the same setting.
> We will include a comprehensive comparison, along with detailed analysis of these results, in the revised version of the paper.
>
> **[Q2] Does the theory extend to attention-based GNNs (GAT)?**
>
> This is an excellent and natural follow-up question. The main difference is that in GAT, the adjacency matrix is feature-dependent. In our work, the derived theorem and proofs mostly rely on the Taylor expansion of the activation $\phi$ and the stability analysis of the message-passing fixed points $ x^* $. Therefore, we believe the stability analysis around this fixed point can be carried through, as it would primarily involve Taylor expanding the terms carried by the dynamic adjacency matrix.
>
> $$A(x^{\ast}+\epsilon) = A(x^{\ast}) + \dots$$
>
> Using perturbation theory on matrices, the fixed operator A will be replaced with a state-dependent operator involving the jacobian of A at the fixed point. We leave this formal theoretical extension to GAT for future work and agree that it could be interesting to investigate this generalization.

---

> > ### Author Rebuttal · Reviewer_fKq6 · 2026-04-04
> >
> > i've read all the author rebuttal, and will keep my rating as weak accept

---

> > > ### Author Response · Authors · 2026-04-05
> > >
> > > We thank the reviewer for acknowledging that their concerns have been fully resolved and would be happy to answer to any additional questions.

---

### Decision · Program_Chairs · 2026-04-30

**Decision:**

Accept (spotlight)

**Comment:**

This paper reframes oversmoothing in GNN as a bifurcation problem. All four reviewers recognized the novelty of the approach, with final scores of 5, 4, 4, 4 and all concerns marked as resolved after rebuttal.
The authors provided additional experiments (GELU/Swish/PairNorm ablations, OGBscale results), corrected notational issues, and committed to improve exposition and releasing code, that **need** to be taken into account in the camera ready version.
I strongly recommend to accept it.